# Seepage from an arctic shallow marine gas hydrate reservoir is insensitive to momentary ocean warming

Wei-Li Hong[1,†], Marta E. Torres[2], JoLynn Carroll[1,3], Antoine Crémière[4,†], Giuliana Panieri[1], Haoyi Yao[1] & Pavel Serov[1]

Arctic gas hydrate reservoirs located in shallow water and proximal to the sediment-water interface are thought to be sensitive to bottom water warming that may trigger gas hydrate dissociation and the release of methane. Here, we evaluate bottom water temperature as a potential driver for hydrate dissociation and methane release from a recently discovered, gas-hydrate-bearing system south of Spitsbergen (Storfjordrenna, ∼380 m water depth). Modelling of the non-steady-state porewater profiles and observations of distinct layers of methane-derived authigenic carbonate nodules in the sediments indicate centurial to millennial methane emissions in the region. Results of temperature modelling suggest limited impact of short-term warming on gas hydrates deeper than a few metres in the sediments. We conclude that the ongoing and past methane emission episodes at the investigated sites are likely due to the episodic ventilation of deep reservoirs rather than warming-induced gas hydrate dissociation in this shallow water seep site.

[1] CAGE—Centre for Arctic Gas Hydrate, Environment and Climate, Department of Geosciences, UiT The Arctic University of Norway, Tromsø N-9037, Norway. [2] CEOAS, Oregon State University, Corvallis 97331, Oregon, USA. [3] Akvaplan-niva AS, Fram Centre, Tromsø N-9296, Norway. [4] Geological Survey of Norway, Trondheim 7491, Norway. † Present addresses: Geological Survey of Norway, 7491 Trondheim, Norway (W.-L.H.); Jet Propulsion Laboratory, California Institute of Technology, Pasadena, California 91109, USA (A.C.). Correspondence and requests for materials should be addressed to W.-L.H. (email: wei-l-hung@uit.no).

Gas hydrate is an ice-like compound that is stable under high pressure and low temperature conditions. Dissociating 1 litre of fully saturated gas hydrate releases 169 of methane under atmospheric pressure[1]. Arctic gas hydrate reservoirs are estimated to hold 100–500 gigatons of carbon[2,3], more than 10% of the carbon in global gas hydrate reservoirs[2]. Current models predict a high potential of Arctic gas hydrate dissociation if bottom water temperatures increase by two degrees during the next century[3]. Indeed, gas hydrate dissociation due to a 1 °C warming of bottom water has been hypothesized to explain hydroacoustic flares observed in water depths shallower than 400 m west of Prins Karls Forland (PKF)[4]. However, the recent recovery of carbonate crusts from PKF points to a longer history of gas venting[5].

Excluding the permafrost area in the Arctic Ocean, gas hydrate recovery has been achieved in water depth at 740 m in the Canadian Beaufort Sea[6] and of 1,200 m water depth at Vestnesa Ridge in Fram Strait[7]. To date, there has been no recovery of gas hydrates in the shelf/slope region of the Arctic, such as PKF[4,5], the shelf area of the Beaufort Sea[8] and the Barents Sea[9,10], regions where increasing bottom water temperatures are thought to have the largest influence on gas hydrate stability. Gas hydrates have only been inferred from the presence of bottom simulating reflectors in seismic data from these areas[8,10]. The hypothesis that Arctic methane seepage is enhanced by warming-triggered gas hydrate dissociation cannot be fully evaluated without direct evidence for the presence of gas hydrates in these warming-sensitive regions.

Here, we present and model the porewater data from a recently discovered shallow water cold seep south of Svalbard. The porewater profiles exhibit concave-up shapes, an indication of an evolving and non-steady-state environment[11–13]. Sulfate profiles are used as a proxy for the activity of anaerobic oxidation of methane (AOM)[14], which in turn responds to methane ascending from deeper sediment towards the sediment-water interface. Using transport-reaction models to simulate the temporal development of the porewater system, we investigate the potential mechanisms leading to the concave-up sulfate profiles, and conclude that these are due to increases in methane flux. The model results also indicate that the timing for the latest methane pulse varies significantly among the investigated sites, suggesting that such events do not respond to regional perturbations such as bottom water warming. Nonetheless, we examine whether bottom water warming can be a plausible mechanism as proposed by previous studies[2–4]. We present evidence to show that short-term warming has limited impact on the gas hydrate stability at the investigated area. Collectively, our results indicate that the ongoing and past methane emission events in this region likely reflect the natural state of a fluid system that is controlled by the state properties of gas reservoirs, the episodic opening of fluid conduits and potential self-sealing by gas hydrate and/or carbonate concretions, as shown in gas hydrate provinces elsewhere[15–18].

## Results

**Description of sediment and porewater profiles.** Here, we describe a group of gas-hydrate-bearing mounds in the slope area south of Svalbard (Storfjordrenna, $\sim$380 m water depth, Fig. 1a and Table 1). The mounds are $\sim$500 m in diameter and extend $\sim$10 m in height above the seafloor. Hydroacoustic imaging of bubble plumes in the water column, commonly referred to hydroacoustic flares (Fig. 1b), and visual observations of bubble streams rising from the seafloor confirm active methane seepage in this area. We reported the sediment and porewater data from seven gravity cores and one multi core recovered during two expeditions in May and October 2015 (Fig. 1c, Table 1 and Supplementary Fig. 1). Core IDs will be abbreviated throughout the text ignoring the cruise number. Gas hydrates were observed in three of the cores (911GC, 912GC and 1520GC) with the shallow-most recovery at 0.85 m below seafloor (mbsf). We therefore term these mound-like structures, 'Gas Hydrate Mounds (GHMs)'. Microfractures, commonly attributed to gas expansion during core recovery, were observed in the three cores with gas hydrate as well as in cores 940GC and 1521GC (Supplementary Fig. 1). Basic information and the available data from these sediment cores can be found in Table 1. In five of the cores, we observed discrete authigenic carbonate nodules. Their mineralogy and carbon isotopic composition are listed in Supplementary Table 1.

We establish an age model with two [14]C dates of planktonic foraminifera from a background core 1522GC (Supplementary Table 2) and Zr/Rb ratio from X-ray fluorescence (XRF) core scanning for stratigraphic correlation (Fig. 2). Zr/Rb ratio is a proxy for sediment grain size[19], which is not affected by methane-derived diagenesis. The presence of oxidized layers in cores 920GC, 940GC and 1522GC (as shown in Fig. 2 and Supplementary Fig. 1) adds additional constraints to this age model using literature attributions to these changes described in other cores from the region[20]. From the two [14]C dates, we estimate a sedimentation rate of $2.21(\pm 0.009) \times 10^4 \text{ m yr}^{-1}$ between the sediment depths of 0.71 and 2.21 mbsf. On the basis of our stratigraphic correlation, we conclude that, between ca. 9 and 16 kyrBP, at least four of the sites experienced similar sedimentation rates. The short recovery in other four sites precludes correlations with the rest of the cores. However, based on Zr/Rb profiles, we speculate that sediment in these sites (911GC, 912GC, 1521GC and 904MC) is younger than Pleistocene. We also notice that, some of the sites may experience more intensive erosion than the others as their upper sediments (920GC, 1520GC and 1522GC) are apparently older relative to the top of 940GC. Notwithstanding, the slightly varied but similar depositional characteristics among the sites exclude the influence of major sedimentation events, such as mass transport deposits (MTDs).

An unusual observation from the porewater profiles is the non-steady-state shape of the $SO_4^{2-}$, $\Sigma HS$, total alkalinity (TA), $Ca^{2+}$ and $Mg^{2+}$ profiles at three of the coring sites from one active GHM (Fig. 3), similar to the 'kink-type' profiles described in Hensen et al.[11]. Above the kinks, the concentrations of these ions show little deviation from bottom seawater values, whereas these solute concentrations increase or decrease rapidly within a narrow depth range below the kinks. Such structure is however absent from the $NH_4^+$ whose concentration shows a gradual increase with no apparent kink in all the coring sites. Porewater data from 920GC, a currently inactive GHM (Fig. 1a), are included to illustrate the smooth profiles typical of a steady-state system (Fig. 3).

## Discussion

There are many published explanations for the presence of non-steady-state porewater profiles in marine sediments worldwide[21]. To find the most plausible explanation for our observations, we simulated five different scenarios with a comprehensive transport-reaction model that considers 15 primary porewater species, seven mineral phases and six redox reactions (Fig. 4; see Methods for modelling details). This model is constrained by the measurements of seven key porewater species ($SO_4^{2-}$, $\Sigma HS$, TA, $Fe^{2+}$, $Ca^{2+}$, $Mg^{2+}$ and $NH_4^+$). The scenarios we considered are: irrigation and seawater intrusion due to biological, physical and hydrological processes[12,22] (Scen1);

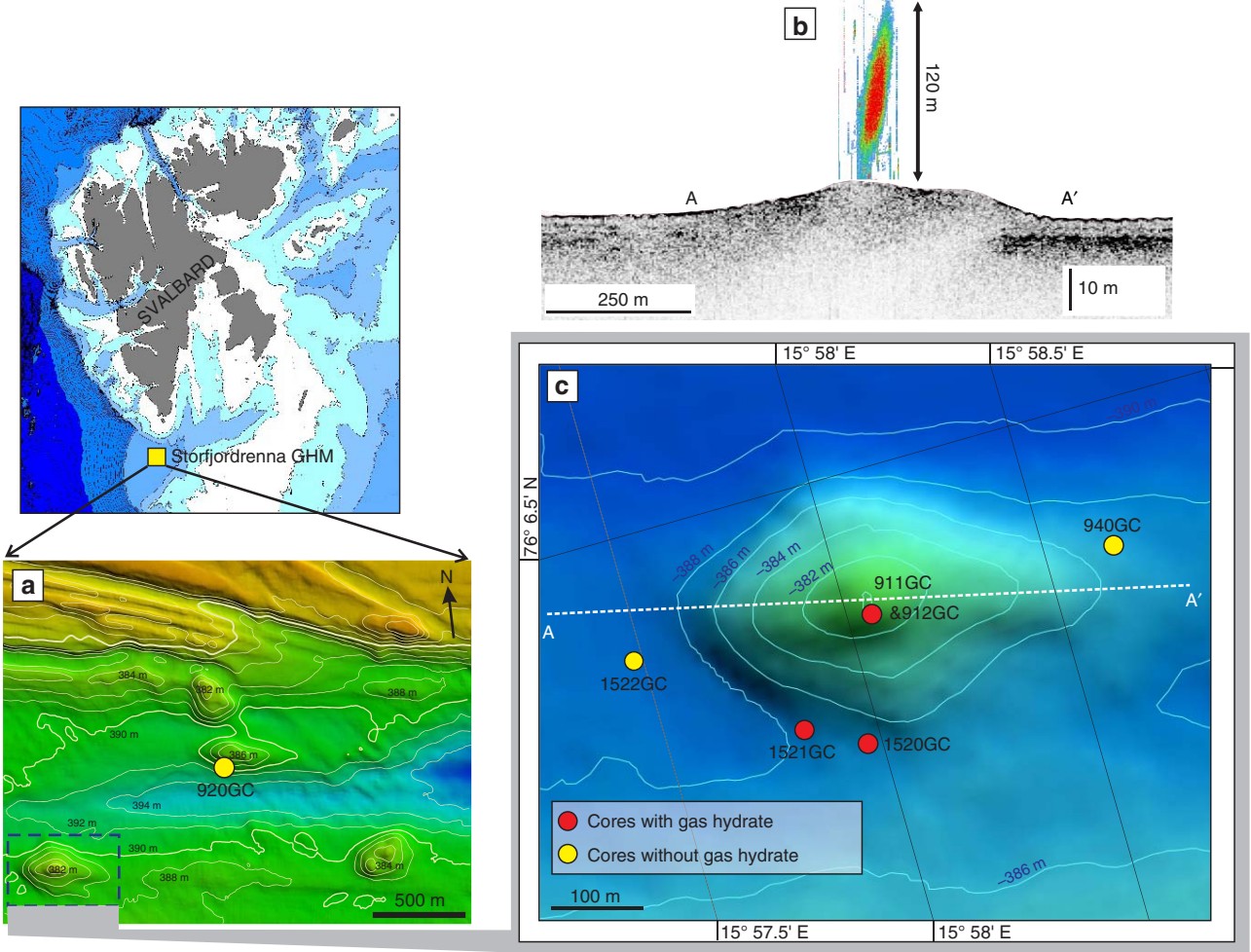

**Figure 1 | Bathymetry and core location from Storfjordrenna gas hydrate mounds.** (**a**) Bathymetry of the Storfjordrenna gas hydrate mounds (GHMs) area. 920GC in **a** shows the location of a coring site with steady-state porewater profiles (Fig. 3). (**b**) The hydroacoustic flare observed only from the summit of the mound. AA' indicates the line of hydroacoustic survey as marked in **c** which shows the detailed bathymetry of the studied GHM. We recovered gas hydrates from three of the study sites (red dots). Notice the different distance scales for the AA' transects in **b,c**.

**Table 1 | Location, water depth and recovery of the eight studied sediment cores.**

| CORE ID | Water depth (m) | Recovery (m) | Lat | Lon | X-ray/XRF | Authigenic carbonate | Porewater |
|---|---|---|---|---|---|---|---|
| CAGE15-2-904MC | 377 | 0.40 | 76.1072 N | 15.9679 E | v | v | v |
| CAGE15-2-911GC | 379 | 0.85 | 76.1069 N | 15.9677 E | v | v | v |
| CAGE15-2-912GC | 380 | 1.04 | 76.1067 N | 15.9686 E | v | NA | NA |
| CAGE15-2-920GC | 386 | 2.50 | 76.1117 N | 16.0108 E | v | NA | v |
| CAGE15-2-940GC | 386 | 3.10 | 76.1069 N | 15.9779 E | v | v | v |
| CAGE15-6-1520GC | 386 | 2.90 | 76.1057 N | 15.9661 E | v | v | v |
| CAGE15-6-1521GC | 386 | 0.95 | 76.1060 N | 15.9638 E | v | v | NS |
| CAGE15-6-1522GC | 388 | 3.20 | 76.1071 N | 15.9579 E | v | ND | NS |

NA, not analysed; ND, not detected; NS, not shown; v, data presented.

changes in sedimentary properties, such as sedimentation rate[11,13] (Scen2) and porosity (Scen3); and changes in methane flux[23] (Scen4). Although strong upwards advection of fluid is unlikely to result in the observed curvatures in our porewater profiles, we still considered a scenario with an advection component to simulate its impact on the porewater profiles (Scen5).

In Scen1, we initiated the model with a shallow SMTZ (grey lines of Scen1 in Fig. 4). Fluid with bottom seawater composition intrudes into surficial sediments by advection, which results in

the observed concave-up sulfate profile (blue lines of Scen1 in Fig. 4). Our model can reproduce the observed sulfate profile with a downwards fluid advection rate of $1\,m\,yr^{-1}$ in 3 months, a similar rate and timescale to what were reported elsewhere[12,24,25]. However, the modelled $NH_4^+$ profile is significantly lower than what is measured due to the dilution from seawater. Such results help us exclude this explanation.

In Scen2, we assumed the top 30 cm of the sediments was initially a layer of MTD with homogenized porewater and sediment composition, which are identical to the composition of

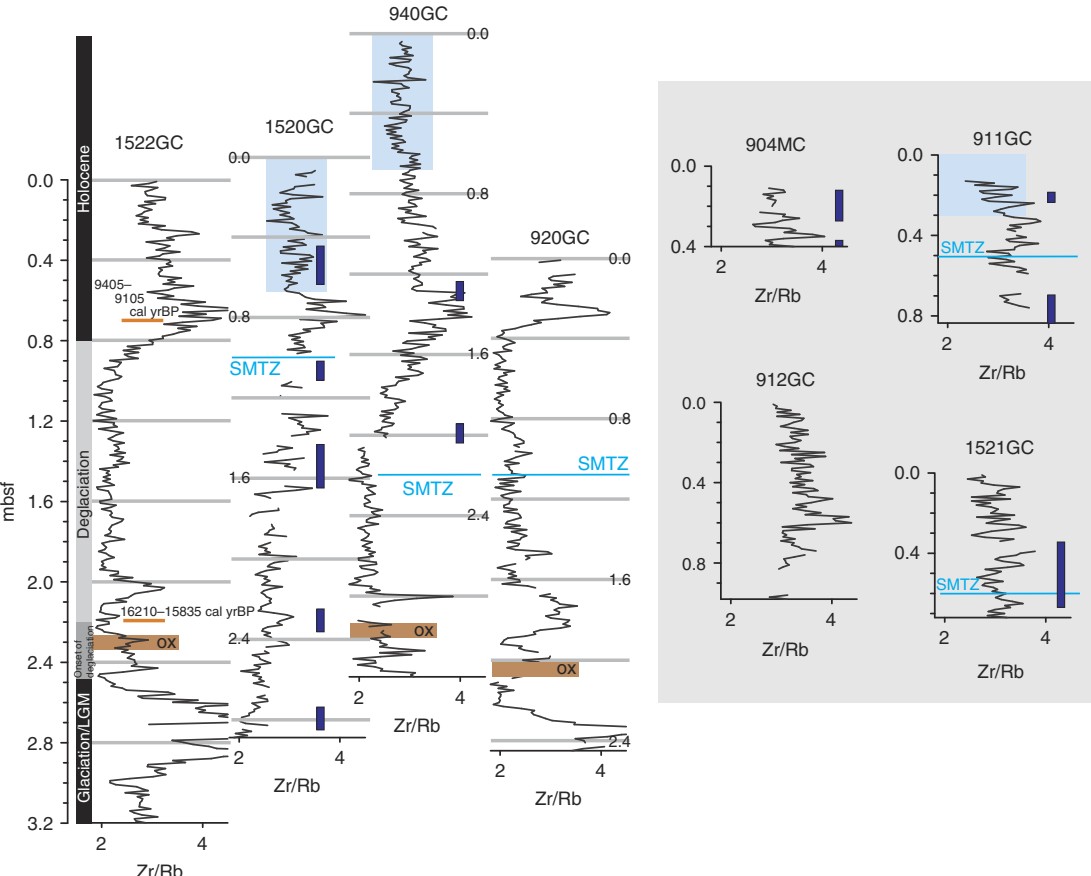

**Figure 2 | Age model for the studied sediment cores.** We compiled Zr/Rb ratio from XRF core scanning, the observations of oxidized layers, and two [14]C dating from planktonic foraminifera to establish the age model for our coring sites. A sedimentation rate of $2.21(\pm 0.009) \times 10^4 \, m \, yr^{-1}$ was estimated between 0.71 and 2.21 mbsf at 1522GC. Oxidized layers observed in cores 920GC, 940GC and 1522GC provide an additional constraint for our stratigraphic correlation. The four cores with <1-m recovery (904MC, 911GC, 912GC and 1521GC), are not well constrained, however, based on the Zr/Rb ratios, we speculate their age to be younger than the Pleistocene. Depths of SMTZ at each core were defined by the sulfate concentration profiles. Approximated depths of authigenic carbonate nodules observed from these cores were indicated by the dark blue bars (see Supplementary Table 1 for values and exact depths). The light blue rectangles covering the Zr/Rb profiles mark the depth ranges of seawater-like porewater.

bottom seawater and seafloor sediments (grey lines in Scen2 of Fig. 4). Six months after the initial condition, diffusion gradually smoothes the profile to the currently observed sulfate profile. With this scenario, we can reproduce most of the porewater profiles but not $Fe^{2+}$, $NH_4^+$ and $\Sigma HS$. As we assumed that the MTD was composed of oxidized sediments with abundant iron hydroxide, the oxidized iron is soon reduced to $Fe^{2+}$ that precipitates as pyrite with hydrogen sulfide, a different scenario from what we have observed. Furthermore, within the 6-month period of simulation, organic matter degradation is not rapid enough to release the observed level of $NH_4^+$. Not only this model scenario fails to explain the porewater profiles, our age model (Fig. 2) indicates no such abrupt sedimentation event.

In Scen3, we evaluated the case with contrasting low porosity in the sediments. We assumed sediments with very low porosity (50%) were deposited for nine centuries. Such deposition results in a 27.5-cm layer of low porosity barrier with the sedimentation rate we assigned. The grey lines of Scen3 in Fig. 4 show the simulation results without such low porosity layer on top, while the blue lines show the results with a low porosity barrier. We observed that such low porosity barrier results in concave-downwards porewater profiles that are different from what we observed. Such porosity contrast is also not expected based on the Cl ratio obtained from XRF core scanning from our sites, which

was used as a proxy for water content[26,27] (Supplementary Fig. 2). These results exclude such explanation for our profiles.

In Scen4, we simulated the case with an increasing methane flux. The simulation results show that an elevated methane flux can deflect the porewater profiles of $SO_4^{2-}, \Sigma HS$, TA, $Fe^{2+}$, $Ca^{2+}$ and $Mg^{2+}$ while not affecting the profile of $NH_4^+$, in agreement with our observations (Fig. 4). We therefore conclude this is the most likely scenario to explain the observed profiles among the four scenarios. Although such model assessment was only performed on the data from 911GC/904MC, we can attribute the same conclusion to 940GC and 1520GC based on the similarity in their porewater profiles (Fig. 3).

In our last scenario (Scen5), we assumed an upwards aqueous advection rate of $1 \, m \, yr^{-1}$ to investigate how the porewater profiles will be impacted. A high methane flux was also assigned to this scenario. To fit the measured $NH_4^+$ profile, we assigned a lower concentration for the initial condition of $NH_4^+$. We observed a 'S-shaped' sulfate profile due to both the high methane flux and advection rate. Concaved downwards profiles were observed from $Ca^{2+}$, and $Mg^{2+}$ (only $Ca^{2+}$ profile is shown), which are different from the measured profiles. By adding such advective component to our model, we are not able to fit most of our porewater profiles. We therefore conclude that aqueous advection is a less significant process compared to diffusion at our study sites and cannot explain our observations in porewater profiles.

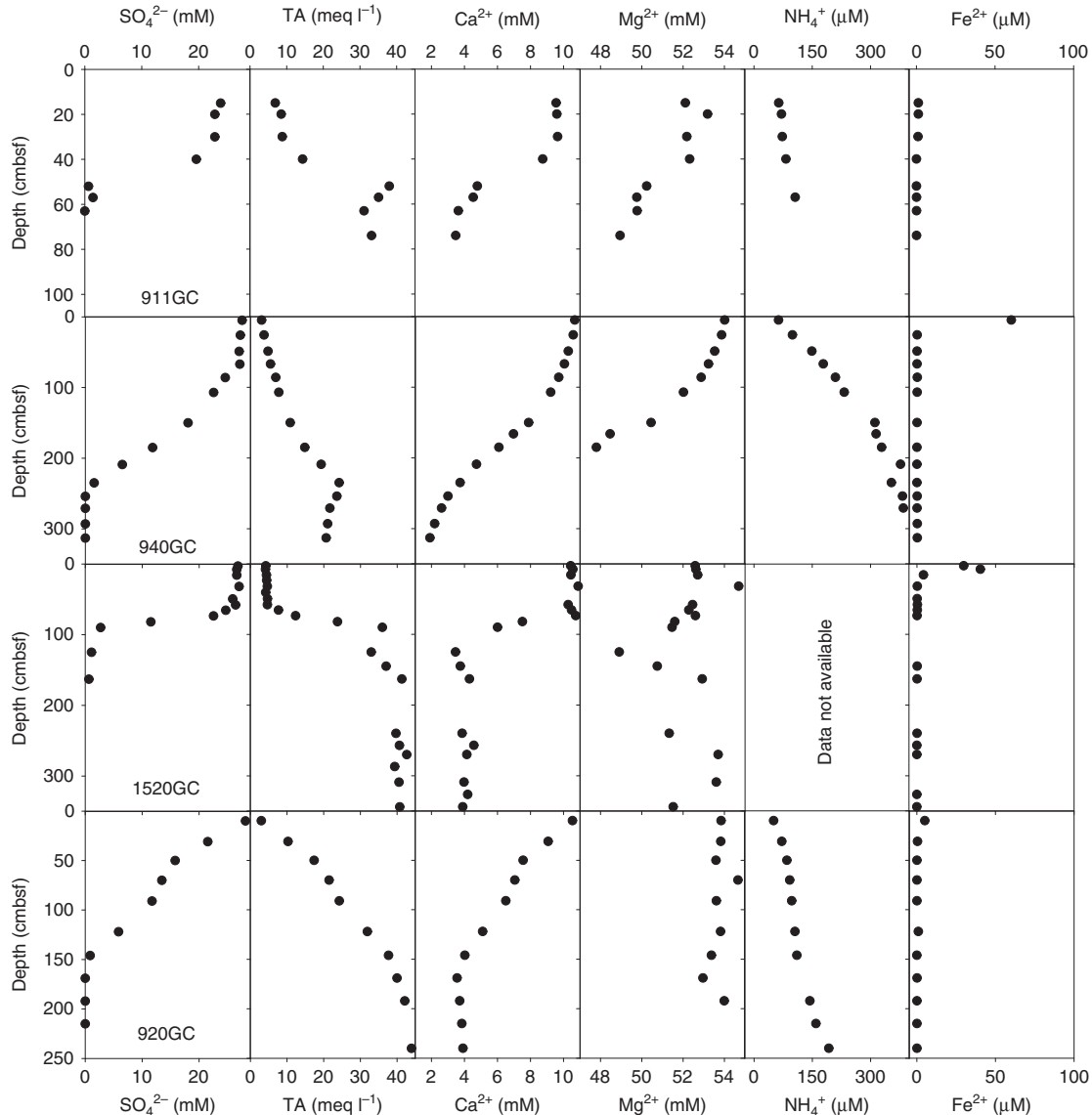

**Figure 3 | Porewater data from four selected cores.** The profiles of sulfate, ΣHS, TA, calcium and magnesium reveal a system out of steady state at three of the sites. Ammonium concentrations at these three sites show smooth profiles without any apparent kinks. We include porewater data from core 920GC to illustrate a steady-state system in the vicinity of the investigated active gas hydrate mound.

As a step further, we aim to estimate the relative timing of changes in methane flux as this information will be valuable for determining its triggering mechanism. If the flux of methane increases at the same time across the investigated GHM, then the system must be responding to regional forcing, such as bottom seawater warming-triggered gas hydrate dissociation. On the other hand, if the timing of methane pulses varies among the investigated sites that are only a few hundred metres apart, then we can conclude that the triggering mechanism must have high geographical heterogeneity.

To estimate the timing of the methane pulses, we simulated the evolution of sulfate profiles at the three coring sites with non-steady-state profiles as they evolve from an initial steady-state situation. As this simulation has to include the entire sediment column above the gas hydrate stability zone (GHSZ), it is computationally too challenging to implement our comprehensive model. We therefore use a reduced model that focuses exclusively on sulfate. This reduced model assumes that sulfate profiles above the kinks are relics of the profiles when the methane supply was weak, for example, 0–0.65 mbsf at 1520GC

(Fig. 3). We derived the initial conditions for each site by executing the same reduced model and adjusting the methane supply from the base of the GHSZ until the results fit the shallow part of the sulfate profile (see Methods section for details). We assume a purely diffusional porewater system with AOM as the only reaction.

By matching our simulations with the observed sulfate profiles, we find that the latest increase in methane supply at the GHM summit was initiated fairly recent (160–340 years at 911GC), and is later than the pulses at its southern (290–630 years at 1520GC) and eastern (1900–4100 years at 940GC) flanks (Fig. 5). From the modelled methane concentration profiles, we can also infer the depths where methane concentration exceeds its saturation in porewater (as calculated by CSMGem[1]). These depths, as shown in Fig. 5, are above the shallowest occurrence of gas hydrates (from cores 911GC to 1520GC) and correspond to the shallowest depth where microfractures were observed in the sediments (Supplementary Fig. 1), a feature that reflects gas expansion during core recovery of sediments with high methane content. Such results suggest that not only does the currently observed

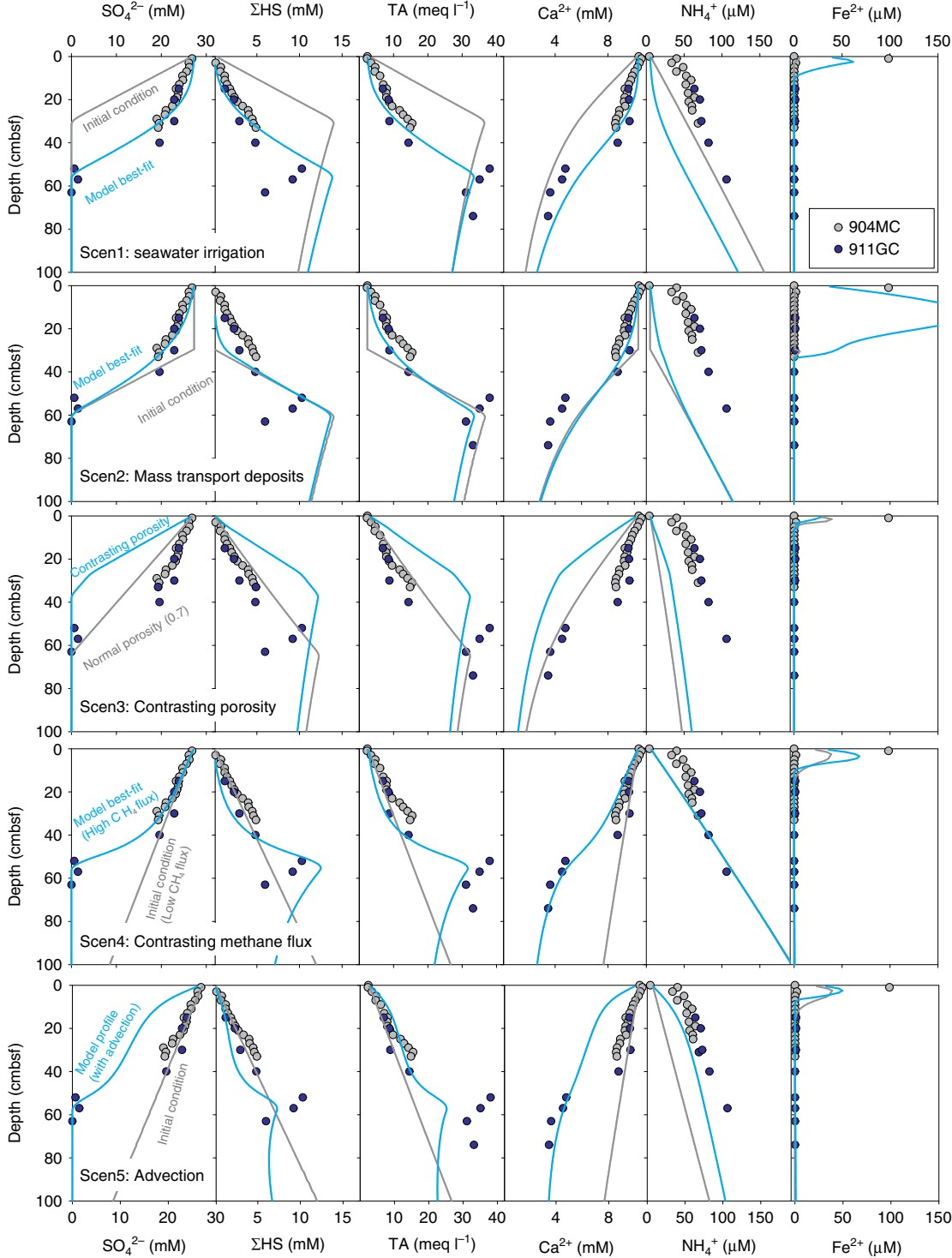

**Figure 4 | Simulation results from five scenarios that may explain the non-steady-state profiles.** We applied a comprehensive model to investigate the cause of the non-steady-state profiles. The first two scenarios (Scen1 and Scen2) can reproduce the kinks in some porewater data but fail to fit the ammonium and iron profiles. Scen3 is dedicated to simulate the porewater profiles with significant porosity contrast which is not expected. The model results fail to fit the observed profiles. Only with an increase in methane flux (Scen4), we can fit all porewater profiles with the model. Aqueous advection (Scen5) will result in concave-downwards profiles for sulfate, calcium and magnesium, which were not observed. We therefore conclude that advection is not significant at the investigated sites.

Storfjordrenna GHM seepage began before the onset of the Anthropocene but the seepage timing differs by orders of magnitude among sites located only a few hundred metres apart. Such results point to triggering mechanisms that are heterogeneous in space and may operate over geological timescales.

Our time estimates may be compromised by not including an advective component in the model. Advection will accelerate the ascending of methane and shorten the time required to achieve the observed concave-up sulfate profiles. As the measured porewater profiles do not correspond to those simulated with

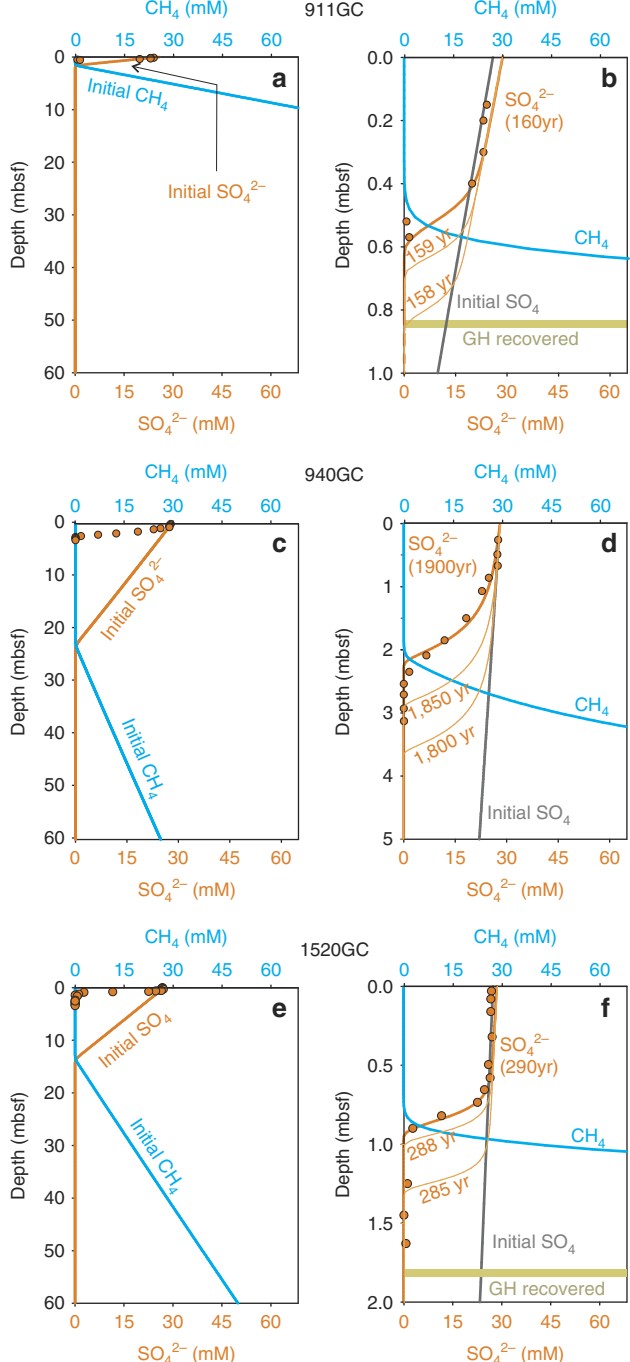

**Figure 5 | Simulation of non-steady-state porewater profiles for the timing of methane seepages.** We applied the reduced model on 911GC (**a,b**), 940GC (**c,d**) and 1520GC (**e,f**) to estimate the timing of seepage. Orange lines in **a**, (**c,e**) are initial sulfate concentrations used in the model which are identical to the black lines in **b**, (**d,f**) for the corresponding depth. These initial conditions were constrained by the shallow part of the measured sulfate profiles (orange dots in all panels) where concentration gradients are small. Blue lines in **a**, (**c,e**) are initial profiles for methane whereas the blue lines in **b**, (**d,f**) are model output for methane, which were also constrained by the first appearance of gas hydrates (yellow bars). Evolution of the modelled sulfate profiles (tortuosity equals to 1.5) at each site were presented in **b**, (**d,f**). The rate of evolution largely depends on the rate of sulfate consumption through anaerobic oxidation of methane, which is fastest at 911GC and slowest at 940GC. Model results reflect differences in the timescales of methane seepage among the three coring sites that are in geographical vicinity.

advection (Scen5 in Fig. 4), we conclude that whereas methane gas is clearly migrating upwards, there is no significant component of aqueous advection at our sites. This is consistent with the observation that the solute profiles in sites with gas discharge (911GC) and those without any evidence of active bubbling (for example, 940GC and 1520GC) can both be simulated with a diffusion-based approach. Decoupling between a gas phase transport dominated by advection concurrent with solute distributions that are dominated by diffusion have been documented in other gas hydrate systems[28]. Our time estimates based on the assumption of a solute diffusion is therefore reasonable.

From our time estimates based on porewater simulations, we infer that the triggering mechanism for a methane flux increase cannot be explained by warming-induced gas hydrate dissociation due to the contrasting timescales within a small region. Nonetheless, as ocean warming has been postulated to drive gas hydrate dissociation along the Svalbard slope region[4], we simulated the propagation of bottom water temperature into the sediments. We aim to elucidate both the depth and the time duration of sediment exposure to temperatures exceeding the gas hydrate phase boundary (see Methods section for model details).

For the past century, the Storfjordrenna area experienced seasonal fluctuations in bottom water temperature from −1.8 to 4.6 °C with occasional anomalies up to 5.5 °C (Fig. 6a, data from World Ocean Database[29]; see Supplementary Table 3 for different data choosing criteria) in response to the dynamic interaction between the warmer Atlantic water and colder Arctic water along the polar front[30]. No apparent warming is observed from Fig. 6a although we acknowledge the scarcity of temperature measurements in the past two decades. The bottom water temperature we measured during the two cruises in May and October 2015 lies within the historical values indicating no obvious temperature anomalies during the months when the studied cores were recovered.

Assuming a sinusoidal fluctuation in seasonal bottom water temperatures over the observed temperature range, our seafloor heat propagation model shows that seasonal temperature fluctuations only affect the gas hydrate shallower than 1.65 mbsf (Fig. 6h). In May, sub-bottom temperatures above 1.1 mbsf are above the temperature threshold for gas hydrate stability (Fig. 6f) while, in October, gas hydrate is within stability field for the entire sediment column (Fig. 6k). Gas hydrates were recovered from 0.85 (911GC) to 2.9 mbsf (1520GC) during these 2 months suggesting that gas hydrate dynamics do not respond quickly to the seasonal temperature fluctuations. Furthermore, our exercise shows that gas hydrates present in sediments deeper than 1.65 mbsf will stay within the stability field throughout the year. This observation again rules out warming-induced gas hydrate dissociation as the cause for the methane flux increase at the investigated coring sites, since our simulations show that the methane pulse responsible for the observed non-steady-state sulfate profiles originates below the current SMZT (0.5–2.2 mbsf, Fig. 3).

We also examined the temperature sensitivity of the system with two different warming trends (Fig. 7). In this model exercise, in addition to the sinusoidal fluctuations in seasonal temperature, a steady increase in mean temperature was assigned to account for the warming in bottom water temperature. We assume an annual warming of 0.033 °C for 30 years in our fast warming case[4] (Fig. 7a). By comparing the assigned temperature fluctuations with the compiled temperature data between 1951 and 1981, the assigned temperatures in summer are comparable to the record temperature for the first decade but are ~1–2 °C higher than the record temperature after ca. 1965. The model results show that even with such fast warming, most of the

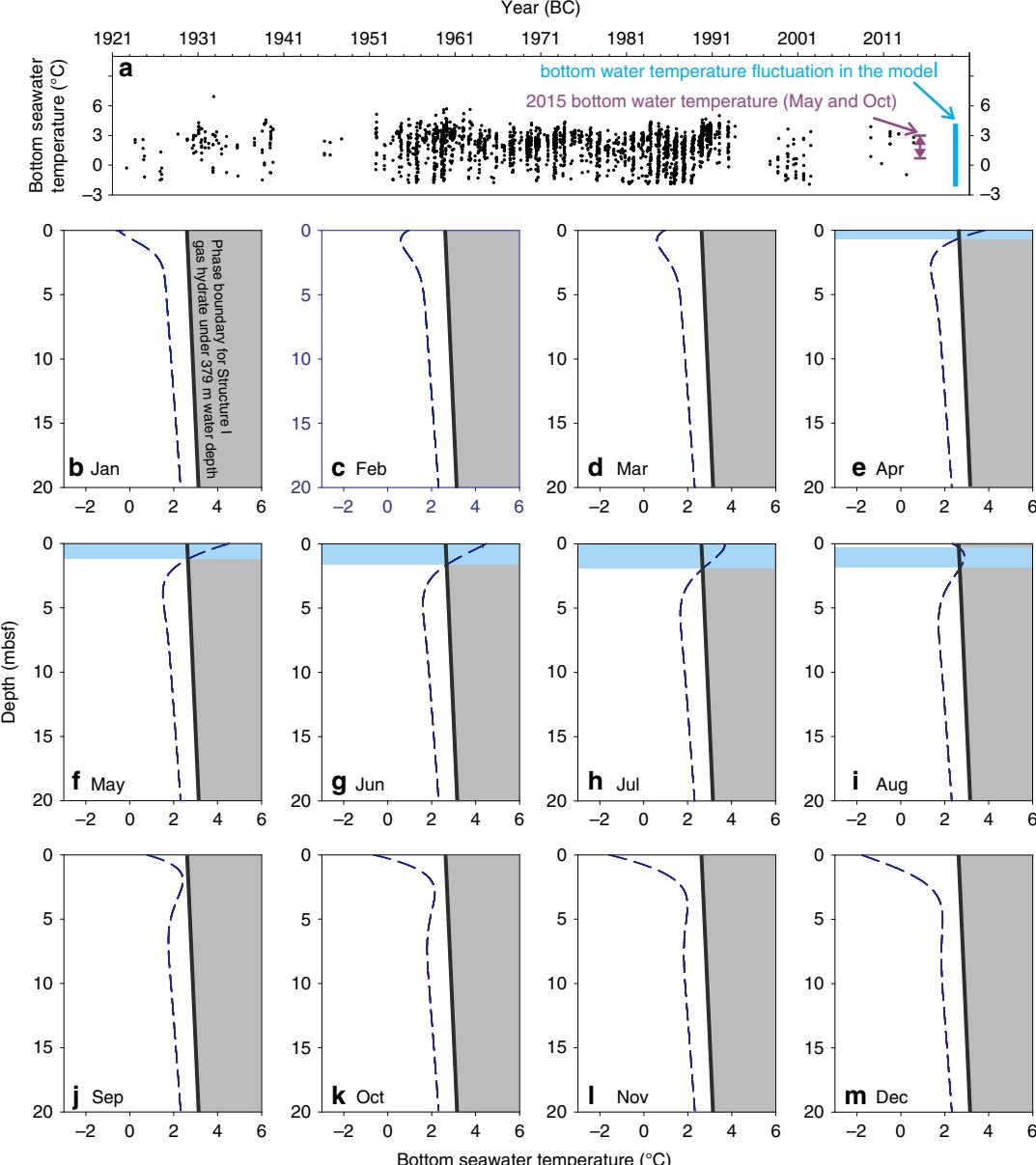

**Figure 6 | Simulation of bottom seawater temperature propagation.** Results of the seasonal heat propagation model with historic bottom water temperatures. (**a**) The average temperature in this area is 1.25 °C with fluctuations between − 1.5 and 5 °C. (**b–m**) Monthly sub-bottom temperature profiles from our model. Seasonal bottom water temperature fluctuations only disturb the sub-bottom gas hydrate shallower than ∼1.65 mbsf (blue area from **e–i**). Our recovery of gas hydrate in May and October, however suggest gas hydrate dynamics do not respond quickly to the seasonal temperature fluctuations.

sediments are still within hydrate stability field except for the top 2.3 m (Fig. 7e) over the 30-year simulation. In the case of slower warming (0.005 °C yr$^{-1}$ for 300 years, Fig. 7f), the sub-bottom temperature for the entire sediment column can exceed hydrate stability field in two centuries (Fig. 7i). As steady increase in annual temperature over centuries is very unlikely, such estimation only reveal the minimum time required. Results from these scenarios could have been possible in the geological past with a lagging time from decades to centuries after the warming initiated based on our temperature modelling.

We observed several discrete layers of authigenic Mg-calcite nodules in five of the cores from Storfjordrenna GHMs (Fig. 2 and Supplementary Table 1). Their depleted δ$^{13}$C, ranging from − 22.6 to − 35.4‰, are consistent with anaerobic oxidation of a methane source and are similar to the carbon isotopic signature

of authigenic carbonates from Barents Sea[31]. Accurate dating of their formation using U/Th dating technique is not possible due to extensive clay particle incorporation within the carbonate matrix[31]. Nonetheless, we interpret these carbonates as indicators of a prolonged and episodic methane seepage history at the GHMs. As the bicarbonate ion produced from AOM at the horizon of SMTZ diffuses both upwards and downwards, authigenic carbonates typically form around the prevailing SMTZ. Therefore, carbonates nodules immediately above and below the current depth of SMTZ likely originate from recent episodes of methane discharge. Carbonates nodules recovered deeper in the cores suggest prior seepage events such as those found well below the current SMTZ at site 1520GC (Fig. 2). On the basis of our $^{14}$C dating and stratigraphy correlation (Fig. 2), the host sediments of these deep carbonate nodules can be older

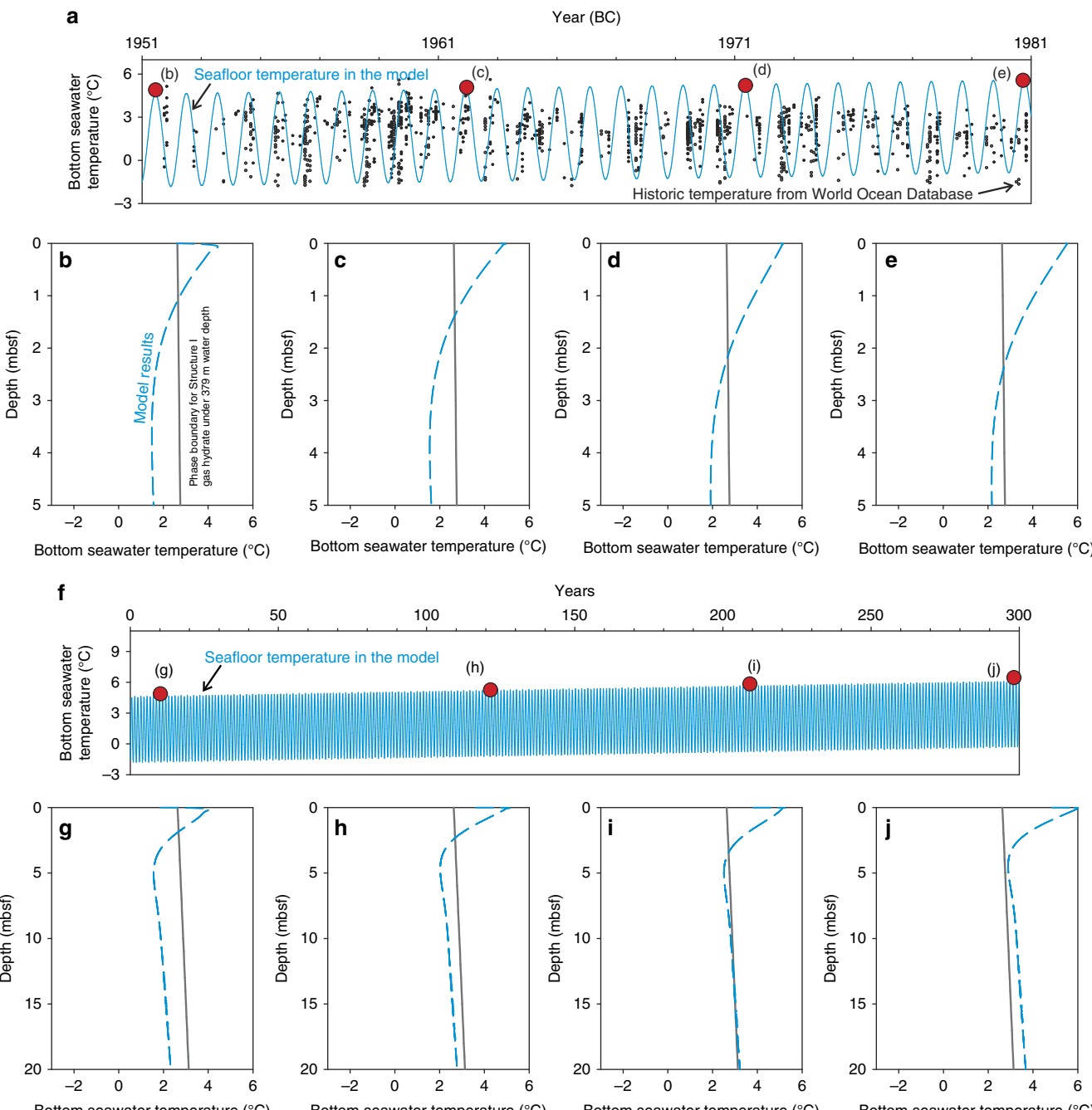

**Figure 7 | Model results of heat propagation with different warming trends.** We simulate the heat propagation from bottom seawater into the sediments with a fast (**a**–**e**) and a slow (**f**–**j**) bottom seawater warming cases. We assumed a 1-degree warming over the course of 30 years for the fast warming case (**a**–**e**) and a 1.5-degree warming in 300 years for the slow warming case (**f**–**j**). Four snapshots of sub-bottom temperature were shown for each case (**b**–**e**) with the time marked on (**a**) as red dots. Our assumed fast warming trend is apparently higher than the historic temperature record especially for summer seasons after *ca.* 1965. Refer to the main text for the inferences of these model results.

than Holocene. We recognize that the sediment age provides only the maximum age constraint for the formation time of the authigenic carbonate nodules. However, the presence of authigenic carbonates deep in the sediment, together with results from our porewater modelling, lead us to propose that the methane discharge in the Storfjordrenna GHMs has been occurring since at least several millenniums ago.

We postulate that the timing heterogeneity and the potentially long history of seepage can be best explained by the natural ventilation of a methane reservoir primarily modulated by pressure conditions at and beneath the GHSZ, and the

opening/sealing of conduits in the sediments. Similar mechanisms for episodic methane venting events have been demonstrated to be plausible by modelling[16] and field observations[17] at other gas-hydrate-bearing margins. Alternatively, it is also likely that gas supply from the reservoir has always been strong but is continuously being redirected due to obstacles in the sediment or near the seafloor. For example, observations from the giant Regab pockmark in the Congo fan, offshore southwestern Africa, documents the development of a natural seal for the methane flow created by the formation of massive authigenic carbonate layers[15]. Future studies are required to differentiate between

these potential drivers for episodic methane discharge at Storfjordrenna GHM.

The recent pursuit by the earth science community to locate areas of methane gas seepage on the seafloor is in part due to the societal concern that warming is accelerating methane leakage at high to mid-latitude regions, thereby potentially forming a feedback scenario for further warming and methane release[7]. Modelling studies that link destabilizing gas hydrate reservoirs to future warming scenarios have augmented this concern[2], lending more urgency to the search for methane bubbles entering the ocean at the seafloor. Contrary to this perspective, our findings, together with other recent studies[5,18,31–33], suggest a long history of methane release, dominantly controlled by large scale Earth system changes (for example, geology, oceanography and glaciology) with gas hydrate as a temporary methane reservoir. The role of gas hydrate should be re-assessed under a more integrated framework by taking each component of the Earth system into consideration[34]. Short-term perturbation from decadal-scale warming of the ocean may have only little consequence to the stability of gas hydrate reservoirs, as our model results suggest. The response and feedbacks between different Earth compartments and methane system[35], whether it is from gas hydrate or not, should receive rather large attention.

## Methods

**Porewater sampling and analyses.** Porewater was sampled at ∼4 °C from both multicores and gravity cores immediately after core recovery using acid-washed rhizon samplers. The samples were collected in 20 ml acid-washed syringes and subsequently filtered through 0.2 μm cellulose acetate in-line filters. Before sub-sampling, the porewater was stored at room temperature for ∼15 min to allow for temperature equilibration. Subsamples were preserved for shorebased analyses of sulfate by adding 6 ml of a 23.8 mM $Zn(OAc)_2$ solution less than 30 min after the syringe was disconnected from the rhizon. Samples for sulfate/sulfide measurements were stored in −20 °C freezer until analysis.

For sulfate analyses, we used a Dionex ICS—1100 Ion Chromatograph outfitted with an AS-DV autosampler and an IonPac AS23 column (eluent: 4.5 mM $Na_2CO_3$/ 0.8 mM $NaHCO_3$, flow: 1 ml min$^{-1}$). The relative standard deviations from repeated measurements of different laboratory standards are better than 0.5% for concentrations above 0.1 mM and better than 1.8% for concentrations above 0.02 mM.

Dissolved iron was determined spectrophotometrically onboard using a ferrospectral complex in ascorbic acid (1%) at wave length of 565 nm. Calibration curves were prepared from iron sulfate standards (10 points from 0.067 to 1 mg l$^{-1}$ $Fe^{2+}$) and determined before each sample batch. Standard and ferrospectral solutions were prepared daily with anoxic 18.2 MΩ MilliQ water using acid-washed volumetric flasks. Measurements were done within an hour after the water samples were extracted.

Concentrations of ΣHS were analysed by the 'Cline method'[36] onshore. Porewater samples fixed with $Zn(OAc)_2$ were well mixed before analyses. Sample (50–200 μl) were diluted to a proper concentration for the analyses. Ten to fifteen minutes after mixing the samples with the colour reagent (N,N-dimethyl-p-phenylenediamine sulfate salt and $FeCl_3 + 6H_2O$ dissolved in cool 18.5% reagent grade HCl), they were measured spectrophotometrically with a wave length of 670 nm. $Na_2S$ standard was made fresh every day before analysing the samples. Thirteen standards with concentrations ranging from 0.04 to 0.25 mM were made for calibration.

TA was measured by Gran titration method a couple hours after the porewater samples were collected onboard. The HCl titrant (0.012 M) was made fresh before the cruise. Before each batch of analyses, 0.01 M borax standard and local seawater were titrated for quality control. Titration was performed in an open beaker with constant stirring. The amount of acid and pH was manually recorded during each acid addition. TA was calculated from the Gran function plots.

Concentrations of calcium and magnesium were measured by the ICP-OES (Leeman Labs Prodigy) in the W.M. Keck Collaboratory for Plasma Spectrometry at the Oregon State University in the radial viewing modes. Samples were diluted 100 times with 1% quartz-distilled nitric acid before analyses. Repeated IAPSO and in-house standard were measured for every 11 samples to assess the instrumental accuracy and precision. Mean concentrations and 1-sigma uncertainties were calculated from three replicate analyses. The uncertainties are generally lower than 1 mmol l$^{-1}$ for magnesium and 0.1 mmol l$^{-1}$ for calcium.

Concentrations of ammonium were determined by a colorimetric method with a Technicon AutoAnalyzer II component at the Oregon State University. The analytical detail is documented in the EPA Criteria 'EPA 600/4-79-020 Methods for Chemical Analysis of Water and Wastes' which is available online[37].

**X-radiograph and X-ray fluorescence scanning.** We scanned all the archived halves of the sediment cores with a GEOTEK X-ray core imaging system (MSCL-

XCT 3.0) at UiT the Arctic University of Norway, using an X-ray intensity of 120 kV and a measuring resolution of 10 mm. Once we identified irregular blocks of higher density relative to the adjacent sediments from the x-radiograph, we then tested these irregular blocks with 2% HCl to confirm their calcareous nature. XRF scanning of the cores was done using the Avaatech instrumentation at UiT. Zr and Rb were quantified with 30 kV, 2,000 μA, at 10 s using Pd filter.

**Mineralogy and stable carbon isotopes of carbonates.** Carbonate samples were powdered and homogenized. Mineralogical analyses were performed by X-ray diffraction on un-oriented samples scanned by a Bruker D8 Advance dif-fractometer (Cu $K_\alpha$ radiation in 3–75° 2θ range). Quantitative data were obtained with the Rietveld algorithm-based code, Topas-4, provided by Bruker. Following a displacement correction of the spectrum made on the main quartz peak, the d$_{104}$ displacement of calcite was used to estimate the $MgCO_3$ in mol% (ref. 38).

An aliquot of the powder prepared for X-ray diffraction was used for stable carbon and oxygen isotopic measurements using a GasBench II preparation line connected to a Thermo Scientific Delta V Advantage IRMS (Thermo Fisher Scientific). Carbon dioxide was produced by the reaction of the powdered sample with 103–105% concentrated phosphoric acid at 70 °C over 2 h. Reproducibility is better than ± 0.15‰ for $\delta^{13}C$ values. Stable isotopic compositions are reported in conventional delta (δ) units relative to Vienna Pee Dee Belemnite reference. Values are reported in Supplementary Table 1.

**$^{14}$C dating.** About 2 mg of planktonic foraminifera (*Neogloboquadrina pachy-derma*) was picked for each depth of $^{14}$C dating. Their $\delta^{13}C$ values were deter-mined at the Stable Isotope Laboratory in the Arctic University of Norway in Tromsø. before dating to ensure no influence of ancient carbon from methane precipitated as secondary carbonate overgrowth or replace the original shells of the foraminifera (Supplementary Table 2). Samples were sent to Beta Analytical for analyses. Both the radiocarbon ages and ages calibrated for local reservoir effect[39] were reported.

**Phase boundary of structure I gas hydrate.** We used CSMGem[1] to estimate the methane solubility (that is, maximum dissolved methane concentration with coexisting hydrate) and the thermodynamic equilibrium temperature of gas hydrate at the base of its stability zone. We assumed Structure I gas hydrate with pure methane and salinity of 35 mg g$^{-1}$. Stability temperature at 60 mbsf, that is, at base of GHSZ, was estimated to be 4.05 °C. We obtained a saturation value of 64 mM for average bottom water temperature (1.25 °C) and pressure (3.85 MPa) conditions. We assume this value is applicable to the shallow (∼1 mbsf) gas hydrates we observed.

**1-D transport-reaction model for porewater profiles.** Two types of modelling were applied on porewater profiles in this work: a comprehensive transport-reac-tion model with full geochemical consideration and a reduced model considering only sulfate and methane. The comprehensive model was applied to investigate the nature of non-steady-state porewater profiles observed from three of the coring sites, whereas the reduced model was applied to estimate the timing of intensified methane flux at these sites.

For the comprehensive model, we coupled a FORTRAN routine, CrunchFlow[40], with a custom MATLAB routine to simulate different biological, hydrological and geological processes that impact the porewater geochemistry. The strategy of coupling CrunchFlow with our custom MATLAB code has been proven successful in our previous work[12]. We used the porewater data from 911GC and 904MC (see Fig. 1 in the main text for location) as the constraints for deciding which scenario is most likely. A perfect fit with the observed porewater profiles is not necessary. Rather, the model should reproduce the main structure of the profiles which are: (1) bended sulfate, ΣHS, calcium, magnesium and TA profiles; (2) High $Fe^{2+}$ only at the top cm; (3) elevated ammonium concentration throughout the core with no apparent kink as in other profiles.

Detailed mathematical formulation of reactions can be found elsewhere[41,42]. We used 12 primary and 5 secondary porewater species in the model (Supplementary Table 4). The primary and secondary species are bounded together through acid–base reactions, which also provide pH buffer to the porewater system (Supplementary Table 5). Six water-rock interactions were included to describe the precipitation/dissolution of various authigenic minerals (Supplementary Table 5). Redox pairs and other aqueous reactions are key to the overall reaction network. We included six such reactions in the model. For the redox pairs, we do not force the coupling among any of the pairs. Instead, electron transfer among the various hydrogen species ($H_2$, $H^+$ and $H_2O$) is the common 'currency' among all the redox reactions. For example, AOM is not strictly coupled with sulfate reduction in the model. Rather, molecular water is reduced to dissolve $H_2$ when methane is oxidized to the bicarbonate ion. Whichever reaction consumes dissolved $H_2$ will facilitate AOM. The tendency of all reactions, including water-rock reactions, are defined by the Gibbs free energy of reaction summarized in Supplementary Table 5.

Organic matter degradation is formulated as a two-step process. In the first step, hydrolysis of organic matter turns solid organic matter to glucose ($C_6H_{12}O_6$) (Supplementary Eq. 8). Fermentation then turns glucose into acetate, $H_2$ and

bicarbonate (Supplementary Eq. 1). Acetate fuels both sulfate reduction (Supplementary Eq. 3) and methanogenesis (Supplementary Eq. 6) whereas $H_2 + HCO_3^-$ induces iron reduction (Supplementary Eq. 11), sulfate reduction (Supplementary Eq. 2) and methanogenesis (Supplementary Eq. 5). Both pathways of sulfate reduction are inhibited when $Fe^{2+}$ concentration in the porewater is higher than 0.4 μM. Both methanogenesis pathways are also inhibited when sulfate concentration is higher than 0.4 mM. These inhibition concentrations were derived from our porewater profiles (Supplementary Table 6). When the condition permits, methane is oxidized to bicarbonate ion (Supplementary Eq. 4) which can be precipitated as authigenic carbonates, consuming calcium and magnesium (Supplementary Eq. 7). The sulfide produced from sulfate reduction will eventually form pyrite (Supplementary Eq. 9) and consumes the $Fe^{2+}$ produced from either reduction of labile iron hydroxide (Supplementary Eq. 11) or dissolution of goethite (Supplementary Eq. 10) in the deeper sediments. We assumed a simplified pathway without considering some of the intermediate species during pyrite formation[43].

For the simulation of the five chosen scenarios, different initial conditions were assigned depending on the situation (grey lines for Scen1, Scen2 and Scen4 in Fig. 4). We applied the same top boundary condition to all scenarios, which was derived from the measurements of bottom water above the sites. No flux condition was assigned for the bottom boundary of all ions except for methane. To account for an additional methane source below the current investigated sediment column, methane was artificially generated in the deepest cell, with a governing parameter that controls the amount of methane that can enter the system.

We simulated a 3-m sediment column with 200 cells cover the first 2 m of the sediment and a 1-m cell dedicated to generate the additional methane needed in Scen4. The total simulation time varies among different scenarios: from 0.25 years (Scen1), 0.5 years (Scen2), 0.7 years (Scen5), 2.5 years (Scen4), to 900 years (Scen3). The model time for each case was determined based on the general fitting with the porewater profiles. We assumed a constant porosity (0.7) throughout the core except for Scen3, which has contrastingly low porosity (0.5) sediments depositing from the core top. Porosity was corrected for tortuosity by assigning a formation factor of 1.5 (ref. 44). Diffusion coefficients for ion were computed[45] assuming a constant temperature of 1.3 based on the average seafloor temperature in the area (Fig. 6a). We assumed a constant sedimentation rate for all scenarios ($3.4E-4$ m yr$^{-1}$) based on the average sedimentation in the area[46]. This value is slightly higher than that calculated based on the two $^{14}C$ dating of foraminifera from 1522GC ($2.2E-4$ m yr$^{-1}$, Fig. 2). We used the higher value as it is based on a larger data set, but note that using the lower value based on our two $^{14}C$-dates does not impact our conclusion. Kinetic constants for all reactions were derived either from literature[47] or from data fitting as summarized in Supplementary Table 6.

As we simulated advection (bulk sediment burial in all cases with additional fluid advection in Scen1 and Scen5) apart from diffusion and reaction, the coupling frequency between the two software routines and the time discretization in our advection computation ($\Delta t$) are key parameters determining the numerical convergence of the results. Following the advice by Hong et al.[12], we decided a $\Delta t$ of 0.02 years for the sediment burial in all scenarios, 0.001 years for the $\Delta t$ of fluid advection in Scen1, and 0.05 years for the fluid advection in Scen5.

For our reduced 1-D transport-reaction model, we simulate a 60-m sediment column considering diffusion of dissolved methane and sulfate in addition to the consumption of both species by AOM. We consider only the water phase in our model (that is, no solid and gas phases). The governing equations are:

$$\frac{\partial C}{\partial t} = -\frac{1}{\phi}\frac{\partial F}{\partial x} + R_{AOM} \quad (1)$$

$$F = -\phi D_s \frac{dC}{dx} \quad (2)$$

where $\phi$, $D_s$ and $dC/dx$ are sediment porosity (0.7), diffusion coefficient in porous media, and concentration gradient for the two target species, $t$ is time in years, $x$ is depth in metres below seafloor (mbsf), $C$ is the concentration of porewater species in mol m$^{-3}$ (volume of bulk sediments) and $R_{AOM}$ is the AOM reaction rate in mol m$^{-3}$ yr$^{-1}$. Diffusion coefficients for seawater media ($D$) were calculated with temperature set to be the bottom water values (0.56 °C) measured during CTD casts in May 2015. We estimated 0.0158 and 0.0301 m$^2$ yr$^{-1}$ for the diffusion coefficients of sulfate and methane[45], respectively. No available information for tortuosity ($\theta$) in this area limits the accuracy of our model results. To at least constrain the order of magnitude of our age estimation, we ran the model with tortuosity of 1.5 and 2.2, a range that covers the possible tortuosity for clayey sediments with 0.6–0.7 porosity[44]. These tortuosity values were then used to define diffusion coefficients in porous media ($D_s$) following:

$$D_S = \frac{D}{\theta^2} \quad (3)$$

We derive the initial conditions by progressing the model until sulfate profiles match the shallow part of the profiles at each site (Fig. 5). We use no flux boundary as the lower boundary condition for sulfate. Fixed methane concentrations were assigned at the bottom of the model frame as boundary conditions.

We did not include fluid advection induced by sediment burial and compaction as our comprehensive model results suggest no significant advection in the aqueous phase (Scen5 in Fig. 4). We solved equation (1) numerically by discretizing depth using a centreed forward finite difference scheme and time using an implicit Crank–Nicholson scheme. The depth and time discretization ($dx = 0.025$ m for all three sites; $dt = 0.01$ for 1520GC and 911GC; $dt = 0.025$ for 940GC due to the long modelling time) were determined by running the model with progressively smaller discretization until the results were numerically stable and accurate.

We solved the $R_{AOM}$ term in equation (1) explicitly as:

$$R_{AOM} = R_{AOM}^{max} \frac{C_{SO_4}}{C_{SO_4} + k_{half-SO_4}} \frac{C_{CH_4}}{C_{CH_4} + k_{half-CH_4}} \quad (4)$$

where $k_{half-SO_4}$ and $k_{half-CH_4}$ are the half saturation constants for sulfate[48] (0.5 mol m$^{-3}$) and methane[49,50] (5 mol m$^{-3}$), respectively. $R_{AOM}^{max}$ is the theoretical maximum AOM rate obtained by fitting the sulfate profile (2 mol m$^{-3}$ yr$^{-1}$). The magnitude of this value affects only the shape of profiles close to the SMTZ depth but not the rate of SMTZ migration. We assumed that AOM is the only diagenetic reaction involving sulfate and methane consumption in this first order simulation; sulfate consumption and methane production due to organic matter degradation were assumed to be not significant under the timescale investigated as these reactions are not likely to induce the dramatic change in porewater concentration gradients. There are two freely adjusted parameters in this model: boundary condition for methane concentration and the time since the inferred methane pulse initiated. The magnitude of methane flux required was constrained both by the curvature of the sulfate profiles and depth where gas hydrates and/or gas microfractures first appear. The methane flux has to be large enough to simulate AOM that can outcompete sulfate diffusion from seafloor. A constant methane flux that is too small will result in a sulfate profile that lacks the kink structure as observed. The methane concentration at the depth of first hydrate appearance, which should be equal to methane solubility, is an additional and independent constraint for the modelled methane profile. With methane flux being constrained, we can estimate the duration of the methane pulse required to fit the data. We also considered our results as a conservative estimation as no advection component was included. Additional sensitivity tests can be found in Supplementary Fig. 3.

**1-D transport model for temperature propagation.** We considered the heat propagation in a 60-m sediment column ($dx = 0.025$ m and $dt = 0.05$ year). The governing equations were all identical with the reduced model except for the diffusivity of heat in the bulk sediments which is defined as:

$$\kappa = \frac{\phi\lambda_w + (1-\phi)\lambda_s}{\phi\rho_w C_w + (1-\phi)\rho_s C_s} \quad (5)$$

where $\lambda$ is the thermal conductivity, $\phi$ is porosity, $C$ is the specific heat and $\rho$ is density. The subscript $w$ and $s$ indicate water and dry sediments (assuming quartz). For water and dry sediments, we used 0.56 and 8.05 W m$^{-1}$ K$^{-1}$ for $\lambda$, 4.2 and 0.73 J g$^{-1}$ K$^{-1}$ for $C$, and $1.03-10^6$ and $2.60 \times 10^6$ g m$^{-3}$ for $\rho$ (ref. 51). The resulting heat diffusivity is 33.4273 m$^2$ yr$^{-1}$. We estimate the regional geothermal gradient to be 0.044 °C m$^{-1}$, based on an average seafloor temperature of 1.44 °C and the limit of gas hydrate stability (4.05 °C) calculated from CSMGem. This geothermal gradient is close to previously reported values (0.042–0.067 °C m$^{-1}$ for sites in similar locations and water depth[5,52]).

**Code availability.** The computer code (CrunchFlow input files, database files and MATLAB routines) that support the findings of this study are available by contacting the corresponding author (W.-L.H.).

**Data availability.** The data that support the findings of this study are available from the corresponding author (W.-L.H.) upon reasonable request.

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

## Acknowledgements

The authors would like to acknowledge the captains and crew members onboard *R/V Helmer Hanssen* for coring assistance, Dr Daniel J. Fornari from WHOI for operating the MISO TowCam deep-sea imaging system, and Dr Jochen Knies from the Geological Survey of Norway (NGU) for the assistance on ion chromatography, and Dr Matthias Forwick from UiT for the assistance on XRF. The authors also acknowledge the lab technicians, engineers from the Department of Geology (UiT) and NGU for assistance with the analyses. Torger Grytå supported the preparation of figures. This work was supported by the Research Council of Norway through its Centres of Excellence funding scheme (project number 223259) and NORCRUST (project number 255150) as well as the US Department of Energy (DE-FE0013531). M.E.T. acknowledges additional support through a fellowship from the Hanse Wissenschaftskolleg, Germany.

## Author contributions

W.-L.H. collected porewater samples, performed X-ray and XRF analyses, and developed the simulation for sulfate profiles and heat propagation. M.E.T. contributed porewater sampling, analyses, and the design of porewater modelling. J.C. supported the overall analyses and contributed to the construction of the manuscript. A.C. performed the analyses of sulfate concentration, carbonate mineralogy and stable carbon isotopes. G.P. provided carbonate samples and assist the establishment of age model for the investigated cores. H.Y. prepared the materials for [14]C dating and assist the establishment of age model for the investigated cores. P.S. contributed the bathymetry and chirp data as well as the interpretation of these data. All authors contributed to the discussion of the paper at different stages.

## Additional information

**Competing interests:** The authors declare no competing financial interests.

**DOI: 10.1038/ncomms16126** **OPEN**

# Erratum: Seepage from an arctic shallow marine gas hydrate reservoir is insensitive to momentary ocean warming

Wei-Li Hong, Marta E. Torres, JoLynn Carroll, Antoine Crémière, Giuliana Panieri, Haoyi Yao & Pavel Serov

*Nature Communications* 8:15745 doi: 10.1038/ncomms15745 (2017) Published 7 Jun 2017; Updated 30 Jun 2017

The financial support for this Article was not fully acknowledged. The Acknowledgements should have included the following:

The publication charges for this article have been funded by a grant from the publication fund of UiT The Arctic University of Norway.

