## [Peer Review File · Nature Communications]

Reviewers' comments:

Reviewer #1 (Remarks to the Author):

Hong et al. have worked diligently to document episodes of methane release from seafloor sediments and to explore the nature and causes of such methane releases. The task is a complicated one, carried out by fine scientists, but is confounded by a host of complicating factors that the authors have worked hard to overcome. Assumptions are therefore necessary, but in my opinion the authors do not directly and adequately confront those assumptions, and discuss their consequences. For this reason I would counsel the authors to rewrite their manuscript, packaging its findings in a way that clearly explains that they present a possible scenario that explores of the timing and control of such episodic methane releases.

The subject of the manuscript is an important one that grapples with the nature, timing, and causes of methane release from seafloor sediments. As a potent greenhouse gas and integral part of the global carbon cycle, methane contributes to the climate of Earth and carbon cycling. Moreover, Hong et al. address the consequences of a warming Earth upon the release of methane from gas hydrate, a major methane reservoir that is particularly sensitive to climatic changes in subpolar regions like the Arctic of Svalbard. Their approach is novel, but unfortunately its basis and subsequent findings is not supported strongly by available evidence.

This reviewer is in the awkward position of agreeing with the paper's overall hypothesis – subsurface control of upward methane migration rather than surface or near-surface thermal control – but not agreeing with the central assumption of its approach. The modeled scenario of methane releases occurring episodically during the last 160 to 4100 years is certainly possible, but the central assumption is weak. I suggest that the author's re-tool their manuscript and encapsulate within it less certitude and present their very worthy work as an exploration or thought experiment, perhaps changing their title to a question.

The entire approach rests on a single assumption - that the sulfate profiles were approximately linear within the most recent past at the modeled sites, and accordingly that at present the portion of shallowest sulfate data points can be linearly extrapolated to the former location of the sulfate methane transition zone (SMTZ), as shown in Figure 3. The authors write (line 95ff): "...we estimate the initial sulfate-methane-transition-zone (SMTZ) to be at the depth where the extrapolations of such relic sulfate profiles intercept the y-axis in Figure 3." However, they do not mention exactly how they extrapolate the shallow sulfate profiles to depth. Looks like a linear extrapolation to me – why not say so directly?

I would like to see documentation of linear sulfate profiles in the proximity of the study site, outside the footprint of the mound, but no such evidence is presented. Unfortunately, additional cores (shown in extended figure 1) do not show linearity in their sulfate profiles, or they do not penetrate the SMTZ. The sulfate profile of core 1521GC, located off the mound, goes to the SMTZ but is definitely not linear; core 1522GC is not deep enough for such assessment. Linear sulfate profiles in the vicinity would document an initial steady-state condition between upward methane flux and downward sulfate flux as balanced by AOM at the SMTZ. These conditions and linear sulfate profiles are not uncommon in areas where the subsurface methane reservoir is significant and where gas hydrates occur, but these conditions are often the statistical exception rather than the rule. In order for the authors to steadfastly make their assumption, there should be evidence that the assumption is *probably* or *plausibly* valid, not merely possible. Perhaps these data exist in other studies, published or unpublished?

Let me emphasize that the approach is inspired, but entirely hinges on this assumption that is without any direct evidentiary support. The authors also mention that the presence of microfractures and soupy sediments support

their modeling results. They do so only in constraining present-day methane concentration to be consistent with gas expansion upon core recovery and formation of microfractures, and reaching methane gas saturation so that gas hydrate can form. The main point of the modelling exploration is to outline a timing scenario for increased upward methane transport, and these observations do not constrain that critical part of the modelling, nor address the fundamental assumption of the exercise.

So the authors do a wonderful job in setting the phase boundaries for gaseous methane and gas hydrate formation and definitely show that recovery depths of gas hydrates in the study cores (or the presence of soupy sediments that may indicate dissociated gas hydrate) are consistent with these phase relations at present. Again, this point does not support the timing results of the modeling, indirectly or otherwise. We know where at depth methane occurs at present from the measured sulfate profiles and gas hydrate occurrence fits the current condition.

The sulfate and methane profiles certainly could have changed from linear profiles with increased methane delivery to the SMTZ sometime in the past over the last 10^5 to 10^4 years, but alternatively they could have never been in steady-state with methane bubbling out of the seafloor episodically over these time periods. Features of the mound are certainly youthful as the authors ably point out, but the core coverage and age framework seem insufficient to constrain the age of the mound independent of model results.

Moreover, the authors also do not explicitly point out that if their assumption is not valid, then the modelled methane concentration profiles and the time periods necessary to match measured sulfate concentration data are *invalid*. The authors should explicitly state and discuss this caveat. Hence my thought about re-tooling the approach of the manuscript, which should indeed be published.

Another key flaw is that the manuscript does not contain an explicit paragraph that outlines their methods and approach, including their key assumptions. The reader sorely needs a roadmap for the arguments presented in the paper, and the paragraph starting on line 62 should be re-written to do this. Unfortunately, the reader must piece together the approach of the paper.

In my view the manuscript has these additional problems:

- A** A smaller point...I am bothered by the term “gas hydrate pingo”. Pingos on land are ice-cored; no evidence that mounds here are gas-hydrate-cored, merely that gas hydrate occur in small amounts within the sediment. No evidence that these mounds are caused by gas hydrate formation. No discussion or references are given to justify this nomenclature. I would use the descriptive term “mound” and excise any reference to “pingo”. The scenario pictured in Fig. 5 doesn’t have the necessary core coverage and age framework to support it, although it certainly is possible. Lastly, I doubt any buoyancy effect from gas hydrate given the small amount of hydrate recovered in the cores and the amount of sediment overburden??

- B** The model-generated methane profiles for the present day are not constrained by any methane data. The authors mention that it is “difficult” to measure accurate, *in situ* methane concentrations and that is certainly true to a point. Sediment methane measurements will be accurate until methane saturation is achieved when methane enters the gaseous phase and can be lost during core recovery. Any such methane measurements before recovery saturation would be very valuable, as present-day modeled methane concentration profiles would be constrained by these data, adding a test to the veracity of the modeled methane profiles thereafter. As given, model results for methane profiles are unconstrained and perhaps not reliable???

- C** The authors do not clearly state how they use sulfate data to construct methane concentrations (see my point above). On line 273ff, the authors write: “We derive the initial conditions of the model by progressing the model until [modeled] sulfate profiles match the shallow part of the [measured] profiles at each site.” Sorry, as a non-modeler, I have no idea what the passage means so the authors should explain better and further.
- D** The modeled times for sulfate-methane equilibration are unconstrained by firm age information and therefore cannot be properly assessed. The top of the Holocene is perhaps/likely recognized by plankton fossils as the authors present. The horizon OX-1 is a surmised correlation that need not be age-specific. Perhaps ^{14}C ages of organic carbon could be determined? Any absolute dates would be of great utility.
- E** $\delta^{13}\text{C}$ values of authigenic carbonate do suggest carbon contributions from methane and so are related to AOM at or near the SMTZ. Interestingly, the shallowest carbonates occur at or near the SMTZ in cores 940GC and 1520GC; there are authigenic carbonates at the SMTZ in core 911GC as well, but the shallowest occurrence is decimeters above the present-day SMTZ. Also, cores 1522GC, 904MC, 912GC(?), and 1521GC have authigenic carbonates above today’s SMTZ. These occurrences may suggest that methane flux was higher in the recent past to move the SMTZ up to these levels. Carrying that thought a bit further, this occurrence argues against initial linear sulfate profiles and adds a confounding element to the equilibration estimate of 160-340 yr BP for Core 911GC from modelling.

Lastly, please note the typos etc. that I point out below in comments tied to the text.

Comments tied to the text

Line	Comments
1	Title change? “A possible on-off...” Although I agree with the authors’ hypothesis, it is not at all certain.
8	Suggest insert “potentially” – not documented as yet.
14	Should be “non-steady-state” throughout text as it moderates “porewater profiles” in this case.
18	Rhythm has a connotation of periodicity – do the authors want to say this?
22	Header doesn’t fit text; suggest deletion here.
23	Suggest: “... is thought to may promote widespread, sedimentary gas hydrate dissociation...”
23ff	The first several lines of this paragraph are not in logical order? Suggest: (1) Gas hydrate is an ice-like compound containing methane and water, thought to be

common and widespread in Arctic in _____ environments. (2) Dissociating one liter ... (3) Arctic gas hydrate reservoirs..... (4) The pervasive warming...

29 "...increases by ~~two degrees~~ 2°C during...". Must identify temperature scale.

30 1°C

31 Different word than "flare" (?) as there is no light or flame? Plume?

32 Suggest: "... (PKF) (see Fig. 1)." Show Prins Karls Forland plume location on inset map in Figure 1. Add significance to figure caption.

33 Add "likely" or "perhaps" as shown?

40 Suggest adding: "...largest impact with regard to gas hydrate dissociation..."

42 Please be specific; by "geophysical evidence" do you mean BSRs?? Please state.

52 No genetic relationship has been proposed in the manuscript - please explain.

55 Confusing...are the authors quoting the measured temperature range of bottom waters or their range in variation? The authors should quote the measured temp range.

Suggest: "...seabed temperatures fluctuating between 3 and 6°C" If that is indeed the case.

Are "seabed temperature" and bottom-water" temperature the same parameter?

62 Somewhere in this paragraph the authors should explicitly construct a roadmap for the reader that explains their approach, the types of data they will utilize to make their arguments, and describe how measured sulfate data and subsequent modelling was used to construct model methane profiles. (C).

63 Sulfate cannot be a true *proxy* for methane concentration since under steady-state conditions they are almost mutually exclusive and under non-steady-state conditions their concentration and distribution are caused by many factors other than co-consumption by AOM – one can only identify the probable depth (the SMTZ, sulfate-methane transition zone) where methane starts to increase with increasing depth into the sediment.

Suggest: "...we use porewater sulfate profiles to calculate methane concentration profiles, ..."

68 To be very clear about methods, suggest: "we measured sulfate concentration in porewaters from recovered sediment cores to construct sulfate profiles,..."

78 I think the methane "supply" or amount of methane in subsurface reservoirs at this point is largely unknowable(?), I think you are suggesting sulfate profiles are responding to increased methane flux toward the seafloor. Perhaps use "delivery" or "flux" instead of supply?? Perhaps this is merely my personal connotation of the word supply.

Suggest: "...recent increases in upward methane flux toward the seafloor."

80 Sediment fractures formed by gas expansion during core recovery are well documented. How

can the authors distinguish microfractures formed by such artifacts from those formed by *in situ* processes? Perhaps I am confused here...to what do the authors attribute the microfractures?

87ff Porewater Fe concentrations are indeed high for cores 940GC and 1520GC, but extremely low in core 911GC in the upper 20 cm. Perhaps it is more correct to somehow say the portion of the sulfate profile profiles used for model calculations seem unaffected by bioturbation, bioirrigation, etc.

93 This is *the* assumption of the approach and should be mentioned/discussed earlier in the paper.

96 “linear” extrapolations? Just how exactly are the shallow sulfate profiles extrapolated?

97 Suggest: “...increase in upward methane flux [or delivery] from a free gas...”

99 Suggest: “The depth of such a gaseous methane reservoir...”.

102ff Suggest something like: “...cannot keep pace with upward methane delivery so that stoichiometric co-consumption of sulfate and methane by AOM requires shoaling of the SMTZ.”

132 At any of the reported geothermal gradients around Svalbard at other seep sites? Those would be more appropriate for comparison.

137 If gas hydrates are destabilized above 5 mbsf due to bottom-water temperature fluctuations, how can gas hydrate exist at ~0.85, and ~2.3 and ~3.45 mbsf in cores 911GC and 1520GC, respectively. I am confused by the purpose and time duration of the thermal modeling (see also line 533ff below).

The authors should point out the occurrence of gas hydrate within this context and explicitly explain it here.

161 Authigenic carbonate precipitation and equilibrating, non-steady-state sulfate profiles do not require the presence of gaseous methane, only increased flux of dissolved methane. Gas hydrates do require gaseous methane for formation but also can be relict within the cores.

Authigenic carbonates do not *document* gas discharge episodes, but are indeed consistent and plausible with it. Nuance is important.

161ff Perhaps (?) implicit in this discussion of authigenic carbonate formation is the assumption that each horizon of carbonate is age-related. The carbonate layers need only be younger than the host sediment and carbonate-rich horizons formed perhaps through AOM are not necessarily ordered by age stratigraphically within the sediment column.

In other words, “one emission episode” need not have “resulted in the observed layers of authigenic carbonate nodules.” The layers could have formed independently at different times. Indeed, the authors say so – lines 114 to 116.

186 “vice versa” unclear. Suggest: “...regions, thereby potentially forming a feedback scenario for further warming and methane release.”

193 Suggest: “...seafloor methane leakage can be entirely unrelated to anthropogenic activities, contributing methane to Earth’s oceans and atmosphere throughout its history.”

- 194 Suggest insertion of “perhaps” as shown – we don’t know the answer, although I agree with the authors’ point of scale.
- 204 Correct resolution 20 millimeters??? Millimeters (mm) or micrometers (μm)?
- 206 Insert “calcareous” as shown.
- 211 Sorry, what does “micronized” mean? Do you mean broken down into micron-size particles? Powdered?
- 212 “carried out” or “performed”
- 214 “...code, Topas-4, provided by Bruker.”
- 220 Is it really 100% phosphoric acid? Do you mean concentrated or pure?
- 221 Add “values” as shown.
- 242 18 Mega-ohm ($\text{M}\Omega$)?
- 246 Please be specific. Thermodynamic phase boundary of gas hydrate?, dissolved methane concentration?
- 250 Seafloor temperature varies with bottom-water temp, so what temperature value?
- 256-57 Re-write sentence? Don’t understand it.
- 271 Suggest “define”.
- 282ff Sorry, “discretizing” is not a word. What do you mean?? Discrete time steps or increments?
- 296 Suggest: “...AOM is the only diagenetic reaction involving sulfate and methane consumption in this first...”
- 300 The porewater changes are *modeled* – they were not observed.
- 305 Suggest: “A methane flux that is too small will result...”
- 306 What does “too smooth” mean??
- 307 To me, gas hydrate saturation is the amount of gas hydrate contained within the sediments usually expressed as a percentage of the sediment by weight or volume; don’t you mean the *methane saturation*, that is when porewater concentration of methane is sufficient to create a methane bubble, which only then can form gas hydrate.
- Please make this nomenclature change elsewhere in the manuscript.
- 323 quartz

- 502 Figure 1 Excellent figure!
- Show PCF location on inset map and label.
 - Locate plume location with arrow along transect A-A' in 1A, and explain in caption.
- 508 streams
- 511 Figure 2 Excellent figure!
- 518 Suggest: "...were observed in all three cores as shown."
- 521 Figure 3 Good figure!
- Show carbonate nodule and gas hydrate depth locations?
 - Use the same key pictured in Extended Data Figure 4?
- 524 Suggest: "The high dissolved iron content (gray polygons) suggest that the mild sulfate gradients are not due to irrigation processes"
- 528 Insert: "F is the methane flux necessary for modeled sulfate profiles to match the measured sulfate data."
- 531 Figure 4 Figure 4a is very valuable – thanks for including. Refer to comment for Line 55.
- Write out *Structure 1*. There is room!
- 533ff 4b. So sorry, but I am confused by the modeling here. Both minimum (winter) and maximum (summer) bottom water temperatures are modeled to assess their effect on warming pore waters over time. The labels on the curves seem to indicate that both min and max temperatures were tested for their penetrative effects over a duration of just under 30 years?? True?? This approach will give a maximum penetration of heat far beyond that should be experienced seasonally.
- Why not just model real seasonality and mention that you tested for a much longer duration to test sensitivity? I presume the effect would be only centimeters to decimeters deep? Am I missing something here?
- What is the relationship of this modeling and figure to that of Extended Data Figure 2?
- My uncertainty here is related to my comment for line 137.
- 539 Figure 5 What does the lighter layer in panel A represent?
- 542ff The scenario pictured here is perhaps not constrained by enough data.
- Age information insufficient – will ¹⁴C age-dates from organic matter help?
 - Does the amount of gas hydrate as recovered in core really enough to generate give a buoyant affect?

Ext Table 1 Why not write out “Authigenic Carbonate” and “Porewater”? There is room and can delete in legend.

- Add to legend: ^ Present or performed ??
- Suggest: “Most of the study cores are gravity cores (GC) except for multicore 904MC, which was collected by the UiT...”
- Suggest: “In 912GC, although authigenic carbonates are present based on X-ray images...”

Ext Table 2 Add units of °C to table!

- Add asterisk (*) to second line of table as shown;
add footnote: * Encompasses [or includes] Storfjordrenna mound
- Suggest: “...water depth and are within an 8° by 2° area surrounding the Storfjordrenna mound...”
- Suggest: “These measurements occurred from...mostly during spring to...measurements during winter months.”
- Delete last caption line, suggest: “The average and minimum temperature values do not differ much from area to area, but maximum temperatures show slightly higher variation.”

End review

Reviewer #2 (Remarks to the Author):

The manuscript reports on the first discovery of gas hydrate in the shallow parts of the Barents Sea. This in itself is very exciting and worth publication.

The authors then carry on to discuss the relevance of observed carbonate accretions in their sediment cores and conclude that seepage must have been long-term and episodic. From this they conclude that an underlying gas reservoir must release gas episodically and that the observations do not necessarily reflect climate effects. I agree with this conclusion.

The paper is of interest to the community that studies the potential effects of global warming and certainly one of the more relevant pieces of work that I have seen on this subject recently.

As I am a geophysicist I cannot do justice to the geochemical modeling and I would like to ask the editor to rely on other reviewer's assessment of this part of the work.

The paper is well written and illustrated and I have only one major comment and several minor suggestions for improvement.

The major comment is that I would like to see a discussion if the episodicity could be caused by the functioning of the seep system on its own. Gay et al. have documented for a pockmark off Angola that the formation of carbonates leads to a deviation of gas flow. This will cause different pathways of the gas over time and may just signal episodicity while the gas is in fact coming up all the time but in different places. If you just look at three cores from the area you would not pick this up. Such a scenario is supported by your observation of small fractures. Potentially addressing this comment may change the conclusions quite a bit.

The two smaller comments: there is a weird sentence in the first paragraph (l. 58) in which you state that 5-88 % of the Arctic hydrate may dissolve due to global warming. This is just arm-waiving and ignores the fact that most of it would quickly reform hydrate. I would just remove this alarmist sentence as it is not necessary.

Further down (l. 65) you seem to say that you infer methane because you see sulfate. I think this is just an imprecise statement and you don't mean it. This should be clarified.

The list of references is very complete and should be shortened. I thought there was a limit on the number of references for Nature? I noticed one problem here: you give three references for a 2 degree seasonal change of BWT, but none of these is really suitable to make the point. I hope the other references are more precise.

Overall, I enjoyed reading the paper and think it is suitable for publication after minor changes.

Christian Berndt, June 2016

Hong et al. have worked diligently to document episodes of methane release from seafloor sediments and to explore the nature and causes of such methane releases. The task is a complicated one, carried out by fine scientists, but is confounded by a host of complicating factors that the authors have worked hard to overcome. Assumptions are therefore necessary, but in my opinion the authors do not directly and adequately confront those assumptions, and discuss their consequences. For this reason I would counsel the authors to rewrite their manuscript, packaging its findings in a way that clearly explains that they present a possible scenario that explores the timing and control of such episodic methane releases.

We appreciate the reviewer's comment. In this revised manuscript, we follow the reviewer's advice by clearly listing all assumptions in separate paragraphs and providing evidence supporting our assumptions (Line 101-149). When appropriate, we show potential consequences if the assumption were invalid. Because, as the reviewer points out, our assumption about initial condition is critical, we present additional evidence to show that by far the most plausible explanation of the non-steady-state porewater profiles. On the basis of these arguments we investigate the potential consequences of our proposed scenario while reminding the readers that this is not a definite sequence of events, but rather the most likely scenario to explain our observations.

The subject of the manuscript is an important one that grapples with the nature, timing, and causes of methane release from seafloor sediments. As a potent greenhouse gas and integral part of the global carbon cycle, methane contributes to the climate of Earth and carbon cycling. Moreover, Hong et al. address the consequences of a warming Earth upon the release of methane from gas hydrate, a major methane reservoir that is particularly sensitive to climatic changes in subpolar regions like the Arctic of Svalbard. Their approach is novel, but unfortunately its basis and subsequent findings is not supported strongly by available evidence.

This reviewer is in the awkward position of agreeing with the paper's overall hypothesis – subsurface control of upward methane migration rather than surface or near-surface thermal control – but not agreeing with the central assumption of its approach. The modeled scenario of methane releases occurring episodically during the last 160 to 4100 years is certainly possible, but the central assumption is weak.

We have provided additional sensitivity tests and data that strengthens our initial assumptions, as detailed in lines 118-123.

I suggest that the author's re-tool their manuscript and encapsulate within it less certitude and present their very worthy work as an exploration or thought experiment, perhaps changing their title to a question.

We have added discussion on each of the assumptions made in this work and provide additional model results and data, if necessary and available, to support our assumptions. We have also changed the title to reflect this is a possible, not factual, scenario

The entire approach rests on a single assumption - that the sulfate profiles were approximately linear within the most recent past at the modeled sites, and accordingly that at present the portion of shallowest sulfate data points can be linearly extrapolated to the former location of the sulfate methane transition zone (SMTZ), as shown in Figure 3. The authors write (line 95ff): "...we estimate the initial sulfate-methane-transition-zone (SMTZ) to be at the depth where the extrapolations of such relic sulfate profiles intercept the y-axis in Figure 3." However, they do not mention exactly how they extrapolate the shallow sulfate profiles to depth. Looks like a linear extrapolation to me – why not say so directly?

In our new text, we clearly state the assumptions underlying initial conditions, and how they were derived by running the same model with a weaker methane supply so that the resulting sulfate profile fits with the shallow

part of the observed sulfate profile. We also added sensitivity tests to demonstrate the reliability of this assumption (Line 118-123).

I would like to see documentation of linear sulfate profiles in the proximity of the study site, outside the footprint of the mound, but no such evidence is presented. Unfortunately, additional cores (shown in extended figure 1) do not show linearity in their sulfate profiles, or they do not penetrate the SMTZ. The sulfate profile of core 1521GC, located off the mound, goes to the SMTZ but is definitely not linear; core 1522GC is not deep enough for such assessment. Linear sulfate profiles in the vicinity would document an initial steady-state condition between upward methane flux and downward sulfate flux as balanced by AOM at the SMTZ. These conditions and linear sulfate profiles are not uncommon in areas where the subsurface methane reservoir is significant and where gas hydrates occur, but these conditions are often the statistical exception rather than the rule. In order for the authors to steadfastly make their assumption, there should be evidence that the assumption is *probably* or *plausibly* valid, not merely possible. Perhaps these data exist in other studies, published or unpublished?

We included a new sulfate profile from a location ca. 1 km away from investigated sites (920GC, see the new Figure 1 for location). We show that it is possible to have linear sulfate profiles in this area indicating a porewater system in steady-state; the observed profiles are a function of changing methane supply and the life stage of the feature. Although we only show one profile in the revised paper, other three gravity cores recovered during our most recent survey this year all show linear sulfate profiles with the depth of SMTZ similar to the investigated feature (0.7 to 1.5 mbsf, unpublished data). A few sentences have been added to discuss these new results (Line 114-117).

Let me emphasize that the approach is inspired, but entirely hinges on this assumption that is without any direct evidentiary support.

The new sensitivity tests and data from Core 920GC are presented as evidence for the validity of this assumption

The authors also mention that the presence of microfractures and soupy sediments support their modeling results. They do so only in constraining present-day methane concentration to be consistent with gas expansion upon core recovery and formation of microfractures, and reaching methane gas saturation so that gas hydrate can form. The main point of the modelling exploration is to outline a timing scenario for increased upward methane transport, and these observations do not constrain that critical part of the modelling, nor address the fundamental assumption of the exercise.

We agree with the reviewer that the presence of microfractures is not the most robust evidence to constrain the methane content in porewater. Nonetheless, they are still a valid indication of high methane content. We have revised the text and deemphasized the use of microfractures as indication of methane saturation. This observation is still included but as a secondary line of evidence (Line 145-148).

So the authors do a wonderful job in setting the phase boundaries for gaseous methane and gas hydrate formation and definitely show that recovery depths of gas hydrates in the study cores (or the presence of soupy sediments that may indicate dissociated gas hydrate) are consistent with these phase relations at present. Again, this point does not support the timing results of the modeling, indirectly or otherwise. We know where at depth methane occurs at present from the measured sulfate profiles and gas hydrate occurrence fits the current condition.

We do not agree with the reviewer on this point. As we explained in the paper, the timing of the seepage depends on the supply of methane, which in turn determines the dissolved methane concentration and where gas

hydrates were first recovered. If gas hydrate/gas fracture/soupy sediments appear deeper in the sediments, the resulting time estimates can be longer with weaker methane supply.

The sulfate and methane profiles certainly could have changed from linear profiles with increased methane delivery to the SMTZ sometime in the past over the last 10^2 to 10^4 years, but alternatively they could have never been in steady-state with methane bubbling out of the seafloor episodically over these time periods. Features of the mound are certainly youthful as the authors ably point out, but the core coverage and age framework seem insufficient to constrain the age of the mound independent of model results.

We include discussion and new model results to show that, without the initial steady-state, it is difficult to produce such non-steady-state sulfate profiles (Figure 5 in the Extended Data, Line 118-124). Even though we do not have ^{14}C dating results to ping-point the timing of our sediments, we feel confident about our age correlation by using elemental ratios and foraminifera assemblage. Both the oxidized layer observed in our sediments and the characteristic elemental ratios (Ca/Ti and Fe/Ca) have been well documented in other studies in Storfjordrenna (see the literature we cited). We can presume the age of our sediments is older than Holocene.

Moreover, the authors also do not explicitly point out that if their assumption is not valid, then the modelled methane concentration profiles and the time periods necessary to match measured sulfate concentration data are *invalid*. The authors should explicitly state and discuss this caveat. Hence my thought about re-tooling the approach of the manuscript, which should indeed be published.

In the new model results we included in this version of manuscript, we have reassessed our assumption of initial conditions. We have emphasized that our time estimation may only be a minimum value as we assume one single seepage event. If methane supply was interrupted, it will take longer time to arrive the observed porewater profiles as showed by our new model results (Line 131-135).

Another key flaw is that the manuscript does not contain an explicit paragraph that outlines their methods and approach, including their key assumptions. The reader sorely needs a roadmap for the arguments presented in the paper, and the paragraph starting on line 62 should be re-written to do this. Unfortunately, the reader must piece together the approach of the paper.

We have re-structured our manuscript according to the reviewer's suggestion. We clearly stated what has been directly derived from our model (Line 85 to 98) and what are the assumptions involved (Line 99-148).

In my view the manuscript has these additional problems:

- A** A smaller point...I am bothered by the term "gas hydrate pingo". Pingos on land are ice-cored; no evidence that mounds here are gas-hydrate-cored, merely that gas hydrate occur in small amounts within the sediment. No evidence that these mounds are caused by gas hydrate formation. No discussion or references are given to justify this nomenclature. I would use the descriptive term "mound" and excise any reference to "pingo". The scenario pictured in Fig. 5 doesn't have the necessary core coverage and age framework to support it, although it certainly is possible. Lastly, I doubt any buoyancy effect from gas hydrate given the small amount of hydrate recovered in the cores and the amount of sediment overburden??

We have revised the term to gas hydrate mound.

- B** The model-generated methane profiles for the present day are not constrained by any methane data. The

authors mention that it is “difficult” to measure accurate, *in situ* methane concentrations and that is certainly true to a point. Sediment methane measurements will be accurate until methane saturation is achieved when methane enters the gaseous phase and can be lost during core recovery. Any such methane measurements before recovery saturation would be very valuable, as present-day modeled methane concentration profiles would be constrained by these data, adding a test to the veracity of the modeled methane profiles thereafter. As given, model results for methane profiles are unconstrained and perhaps not reliable???

We agree that constraining the modeled methane profiles is tricky as *in-situ* methane concentration is very difficult to obtain. We however do not agree with the reviewer that our modeled methane profiles are totally unconstrained. The first appearance of gas hydrates was 0.85 mbsf at 911GC and 1.8 mbsf at 1520GC (if deduce the huge void due to gas expansion during core recovery below 1.7 mbsf). These are solid constraints of *in-situ* methane concentration as dissolved methane content has to be at saturation (64 mM) with coexisting of hydrates.

The observation of hydrate first appearance is just a conservative estimate as some hydrate may dissociated during core recovery. That is why we also mentioned the observations of gas fractures and soupy sediments in the core. Admittedly, these observations are not as solid as the real presence of hydrates as they can be artifacts of core handling. However, these gas-related features will not be present if methane concentration is too low. The best example would be from 1520GC, such gas expansion fractures do not appear until 1 mbsf. Below such horizon, frequent gas hydrate layers/gas fractures/soupy sediments appear. We consider this is a very strong sign of changes in phases of methane which suggest saturated dissolved methane content below 1 mbsf. We have deemphasized the use of microfractures as a solid evidence for methane saturation but still include such observations as a secondary support for high methane content. A new paragraph has been included to discuss the use of constraints for methane concentrations (Line 136-148).

We include available *ex-situ* methane profiles from some of the investigated cores, which are currently included in another paper that was just submitted (Serov et al.). Since these *ex-situ* data below the SMTZ is not reliable, we don't think adding these data here will in any way strengthen our arguments.

C The authors do not clearly state how they use sulfate data to construct methane concentrations (see my point above). On line 273ff, the authors write: “We derive the initial conditions of the model by progressing the model until [modeled] sulfate profiles match the shallow part of the [measured] profiles at each site.” Sorry, as a non-modeler, I have no idea what the passage means so the authors should explain better and further.

We have clarified this point in the revised paper (Line 111-117).

D The modeled times for sulfate-methane equilibration are unconstrained by firm age information and therefore cannot be properly assessed. The top of the Holocene is perhaps/likely recognized by plankton fossils as the authors present. The horizon OX-1 is a surmised correlation that need not be age-specific. Perhaps ^{14}C ages of organic carbon could be determined? Any absolute dates would be of great utility.

It is difficult to date any of the cores reported here with ^{14}C technique due to the potential overprint from ancient carbon from methane. Our correlation by using elemental ratios and unique paleoceanography events can be more robust as they were not significantly impacted by any methane-induced diagenesis. The characteristic Ca/Ti ratios and oxidized layers (ox) that we observed from our cores were well documented by previous studies in the sediments of Storfjordrenna area (Lucchi et al. 2013). In the revised Extended data, we also added Fe/Ca ratio for all the sites investigated as an additional constraint for correlation. Łacka et al (2015) has shown how this ratio switches from low to high values as the sediments influenced more by the sediments from ice melting before 11ka to ice-free ocean afterward. Combined all the observations we can establish a rough but robust time frame for the study sites, which is sufficient for the argument we made in the paper.

E $\delta^{13}\text{C}$ values of authigenic carbonate do suggest carbon contributions from methane and so are related to

AOM at or near the SMTZ. Interestingly, the shallowest carbonates occur at or near the SMTZ in cores 940GC and 1520GC; there are authigenic carbonates at the SMTZ in core 911GC as well, but the shallowest occurrence is decimeters above the present-day SMTZ. Also, cores 1522GC, 904MC, 912GC(??), and 1521GC have authigenic carbonates above today's SMTZ. These occurrences may suggest that methane flux was higher in the recent past to move the SMTZ up to these levels. Carrying that thought a bit further, this occurrence argues against initial linear sulfate profiles and adds a confounding element to the equilibration estimate of 160-340 yr BP for Core 911GC from modelling.

It is true that there are authigenic carbonates above the present SMTZ, which indicate the methane flux must have been stronger than it is today. However, there are also sediment sections that have no observable carbonates, such as the first 0.1m in GC911, the first 1.2m in GC940, the first 0.4m in GC1520. These sections suggest a period with very weak methane supply. During this quiescent period when methane supply is weak, porewater system may very likely reach a steady state due to the less dynamic seepage activity. We now included this caveat to our discussion chapter (Line 225-228).

Lastly, please note the typos etc. that I point out below in comments tied to the text.

Comments tied to the text

Line	Comments
1	Title change? "A possible on-off..." Although I agree with the authors' hypothesis, it is not at all certain. We have changed the title as suggested.
8 done	Suggest insert "potentially" – not documented as yet.
14	Should be "non-steady-state" throughout text as it moderates "porewater profiles" in this case. We have changed to non-steady-state throughout the paper.
18	Rhythm has a connotation of periodicity – do the authors want to say this? Changed in the text, We have changed rhythm to a different term throughout the paper.
22	Header doesn't fit text; suggest deletion here. We have changed the header to better fit the text.
23	Suggest: "... is thought to may promote widespread, sedimentary gas hydrate dissociation..." We have deleted the sentence.
23ff	The first several lines of this paragraph are not in logical order? Suggest: (1) Gas hydrate is an ice-like compound containing methane and water, thought to be ... (3) Arctic gas hydrate reservoirs..... (4) The pervasive warming... see line 21-26 common and widespread in Arctic in _____ environments. (2) Dissociating one liter
29	"...increases by two degrees 2°C during...". Must identify temperature scale.

In the cited paper, the 2C increase in temperature is a regional phenomenon. The authors applied such change to a broad region with bottom water temperature over a range of 12 degrees. An exact temperature range cannot therefore specified.

30 1°C
Done (line 27)

31 Different word than “flare” (?) as there is no light or flame? Plume?
Hydroacoustic flare is a common term in the literature. We therefore decide to keep the term but add reference to its past use and clarify that these reflect bubbles in the water column (lines 46-47).

32 Suggest: “...(PKF) (see Fig. 1).” Show Prins Karls Forland plume location on inset map in Figure 1. Add significance to figure caption.-
Done, see the new figure 1.

33 Add “likely” or “perhaps” as shown?
The sentence has been deleted.

40 Suggest adding: “...largest impact with regard to gas hydrate dissociation...”
done (line 37-38)

42 Please be specific; by “geophysical evidence” do you mean BSRs?? Please state.
Done (line 38-39)

52 No genetic relationship has been proposed in the manuscript - please explain. Deleted “genetic”
This sentence has been deleted.

55 Confusing...are the authors quoting the measured temperature range of bottom waters or their range in variation? The authors should quote the measured temp range.

Suggest: “...seabed temperatures fluctuating between 3 and 6°C” If that is indeed the case.

Are “seabed temperature” and bottom-water” temperature the same parameter?

We have unified the term throughout the paper.

We have changed the temperature to absolute range (Line 54)

62 Somewhere in this paragraph the authors should explicitly construct a roadmap for the reader that explains their approach, the types of data they will utilize to make their arguments, and describe how measured sulfate data and subsequent modelling was used to construct model methane profiles. (C).

We provide such road map after the introduction paragraphs (Line 60-75).

63 Sulfate cannot be a true *proxy* for methane concentration since under steady-state conditions they are almost mutually exclusive and under non-steady-state conditions their concentration and distribution are caused by many factors other than co-consumption by AOM – one can only identify the probable depth (the SMTZ, sulfate-methane transition zone) where methane starts to increase with increasing depth into the sediment.

We now rephrase that we use sulfate as a proxy for AOM activity which response to methane supply. We believe this a more accurate statement (Line 61-62).

Suggest: “...we use porewater sulfate profiles to calculate methane concentration profiles, ...”

68 To be very clear about methods, suggest: “we measured sulfate concentration in porewaters from recovered sediment cores to construct sulfate profiles,…”

done (line 60)

78 I think the methane “supply” or amount of methane in subsurface reservoirs at this point is largely unknowable(?), I think you are suggesting sulfate profiles are responding to increased methane flux toward the seafloor. Perhaps use “delivery” or “flux” instead of supply?? Perhaps this is merely my personal connotation of the word supply.

Flux is a term with precise definition (material over a certain area and time). Supply, on the other hand, can be applied in a wider range. We therefore stick to our choice of word in this case.

Suggest: “...recent increases in upward methane flux toward the seafloor.”

80 Sediment fractures formed by gas expansion during core recovery are well documented. How can the authors distinguish microfractures formed by such artifacts from those formed by *in situ* processes? Perhaps I am confused here...to what do the authors attribute the microfractures?

We think most of the microfractures observed from the core are artifacts due to gas expansion upon core recovery. They are indication of high methane content but not necessary methane saturation. We have clearly stated this in the revised paper (Line 146-148).

87ff Porewater Fe concentrations are indeed high for cores 940GC and 1520GC, but extremely low in core 911GC in the upper 20 cm. Perhaps it is more correct to somehow say the portion of the sulfate profile profiles used for model calculations seem unaffected by bioturbation, bioirrigation, etc.

There is an ongoing argument in the literature about whether such kinked-shaped sulfate profiles are due to changes in methane flux or irrigation processes (see the references we cited in the paper). The reason why we included the Fe²⁺ measurements is to exclude the irrigation processes. The low Fe²⁺ concentration at the top of 911GC, but still 10 times higher than the concentration in overlying seawater, is due to the sampling resolution of this core. If we have taken porewater samples from the first 5 cm of the core, as we did for 940GC and 1520GC, we would have detected such high ferrous iron concentration.

93 This is *the* assumption of the approach and should be mentioned/discussed earlier in the paper. We have added a new paragraph (Line108-124) to explicitly discuss this assumption.

96 “linear” extrapolations? Just how exactly are the shallow sulfate profiles extrapolated?
We use the sample model to derive those initial conditions. See lines 111-114.

97 Suggest: “...increase in upward methane flux [or delivery] from a free gas...”
See our reply above

99 Suggest: “The depth of such a gaseous methane reservoir...”.
We have modified the statement (line 86-87)

102ff Suggest something like: “...cannot keep pace with upward methane delivery so that stoichiometric co-consumption of sulfate and methane by AOM requires shoaling of the SMTZ.”
Done, see Line 92-94

132 At any of the reported geothermal gradients around Svalbard at other seep sites? Those would be more appropriate for comparison.

There is very little measurements around Svalbard. The Crane et al. paper is one of the most cited work in the area. There are also some geothermal measurements from Berndt et al. (2014) which we also cited in this version of manuscript.

137 If gas hydrates are destabilized above 5 mbsf due to bottom-water temperature fluctuations, how can gas hydrate exist at ~0.85, and ~2.3 and ~3.45 mbsf in cores 911GC and 1520GC, respectively. I am confused by the purpose and tim duration of the thermal modeling (see also line 533ff below).

The authors should point out the occurrence of gas hydrate within this context and explicitly explain it here.

To avoid confusion, we have deleted this sentence. Our model results suggest that high bottom water temperature can propagate up to 15 mbsf during summer and may destabilize the gas hydrate shallower than 5 mbsf. However, we still recover gas hydrate during the two cruises before and after summer. We can show that such seasonal temperature fluctuation has only limited impact on gas hydrate, which may be due to the kinetic of hydrate dissociation and the cold bottom water temperature in winter. The depths where gas hydrate were recovered were noted in Figures 2 and 3.

161 Authigenic carbonate precipitation and equilibrating, non-steady-state sulfate profiles do not require the presence of gaseous methane, only increased flux of dissolved methane. Gas hydrates do require gaseous methane for formation but also can be relict within the cores.

Authigenic carbonates do not *document* gas discharge episodes, but are indeed consistent and plausible with it. Nuance is important.

We have deleted that statement.

161ff Perhaps (?) implicit in this discussion of authigenic carbonate formation is the assumption that each horizon of carbonate is age-related. The carbonate layers need only be younger that the host sediment and carbonate-rich horizons formed perhaps through AOM are not necessarily ordered by age stratigraphically within the sediment column.

In other words, “one emission episode” need not have “resulted in the observed layers of authigenic carbonate nodules.” The layers could have formed independently at different times. Indeed, the authors say so – lines 114 to 116.

We fully aware that there is no age relation between all the layers of carbonate nodules. A statement has been added to clarify that (Line 203-204).

186 “vice versa” unclear. Suggest: “...regions, thereby potentially forming a feedback scenario for further warming and methane release.”

Done (Line 242-243)

193 Suggest: “...seafloor methane leakage can be entirely unrelated to anthropogenic activities, contributing methane to Earth’s oceans and atmosphere throughout its history.”

Done (Line250-251)

194 Suggest insertion of “perhaps” as shown – we don’t know the answer, although I agree with the authors’ point of scale.

Done (Line 251)

204 Correct resolution 20 millimeters??? Millimeters (mm) or micrometers (µm)?

mm is correct. We scan the core with XRF every 10 mm.

206 Insert “calcareous” as shown.

Done (Line 262)

211 Sorry, what does “micronized” mean? Do you mean broken down into micron-size particles?
Powdered?

We have changed to a different word (Line 268).

212 “carried out” or “performed”

Done (Line 269)

214 “...code, Topas-4, provided by Bruker.”

Done (Line 271)

220 Is it really 100% phosphoric acid? Do you mean concentrated or pure?

Concentrated acid, corrected (Line 277)

221 Add “values” as shown.

Done (Line 279)

242 18 Mega-ohm (MΩ)?

Corrected (Line 299)

246 Please be specific. Thermodynamic phase boundary of gas hydrate?, dissolved methane concentration?

Clarified (Line 304-305)

250 Seafloor temperature varies with bottom-water temp, so what temperature value?

Done (Line 308)

256-57 Re-write sentence? Don’t understand it.

Done (Line 315-316)

271 Suggest “define”.

Done (Line 329)

282ff Sorry, “discretizing” is not a word. What do you mean?? Discrete time steps or increments?

“Discretize” is the word used in many modeling and mathematics literatures for the action of transforming continuous space and temporal regime into discrete cells.

296 Suggest: “...AOM is the only diagenetic reaction involving sulfate and methane consumption in this first...”

Done (Line 355-356)

300 The porewater changes are *modeled* – they were not observed.

Deleted (Line 359)

305 Suggest: “A methane flux that is too small will result...”
Done (Line 364)

306 What does “too smooth” mean??
Corrected (Line 364-365)

307 To me, gas hydrate saturation is the amount of gas hydrate contained within the sediments usually expressed as a percentage of the sediment by weight or volume; don’t you mean the *methane saturation*, that is when porewater concentration of methane is sufficient to create a methane bubble, which only then can form gas hydrate.
Corrected (Line 366-367)

Please make this nomenclature change elsewhere in the manuscript.

323 quartz
Corrected (Line 382)

502 Figure 1 Excellent figure!
• Show PKF location on inset map and label.
• Locate plume location with arrow along transect A-A’ in 1A, and explain in caption.

Done

508 streams
Corrected (Line 541)

511 Figure 2 Excellent figure!

518 Suggest: “...were observed in all three cores as shown.”
Done (Line 547)

521 Figure 3 Good figure!
• Show carbonate nodule and gas hydrate depth locations?
Shown by the arrows
• Use the same key pictured in Extended Data Figure 4?

Done

524 Suggest: “The high dissolved iron content (gray polygons) suggest that the mild sulfate gradients are not due to irrigation processes”
Done (Line 549-550)

528 Insert: “F is the methane flux necessary for modeled sulfate profiles to match the measured sulfate data.”

Done (Line 554)

531 Figure 4 Figure 4a is very valuable – thanks for including. Refer to comment for Line 55.
• Write out *Structure 1*. There is room!

Done

533ff 4b. So sorry, but I am confused by the modeling here. Both minimum (winter) and maximum (summer) bottom water temperatures are modeled to assess their effect on warming pore waters over time. The labels on the curves seem to indicate that both min and max temperatures were tested for their penetrative effects over a duration of just under 30 years?? True?? This approach will give a maximum penetration of heat far beyond that should be experienced seasonally.

Why not just model real seasonality and mention that you tested for a much longer duration to test sensitivity? I presume the effect would be only centimeters to decimeters deep? Am I missing something here?

As we explained in the paper, we simulated the seasonal bottom seawater temperature fluctuation (from -1.8 to 4.6 °C) for over 30 years to see how such variation can affect sub-bottom temperature regime and therefore gas hydrate stability. The temperature profile changes according to season but overall does not change over time. As we still recovered gas hydrate in May and October, we can prove that such temperature perturbation does not cause significant gas hydrate dissociation. We have modified the figure to show only the seasonal temperature change within a year and note the profiles are the same after 30 years.

What is the relationship of this modeling and figure to that of Extended Data Figure 2?

In Extended Data Figure 2, we show the results of the model with an extreme scenario with 5 degrees higher bottom water temperature. This had happened in geological past (see our citation) but never seen from our century-long record. We explained this better in the text (Line 179-181).

My uncertainty here is related to my comment for line 137.

539 Figure 5 What does the lighter layer in panel A represent?

That is a thin gas hydrate layer. We have clarified the legend.

542ff The scenario pictured here is perhaps not constrained by enough data.

- Age information insufficient – will ¹⁴C age-dates from organic matter help?
- Does the amount of gas hydrate as recovered in core really enough to generate give a buoyant affect?

As in our previous reply, ¹⁴C dating of the material from our investigated cores is very challenging. We currently have no such information available. Our correlation by using elemental ratios in the sediments and unique paleoceanography events maybe an alternative that provide robust age framework.

We have also changed the figure legend to “volume expansion due to gas hydrate layers” which is more likely than buoyancy as gas hydrate has density 10% less than water. Such volume expansion may partly explain the mound topography.

Ext Table 1 Why not write out “Authigenic Carbonate” and “Porewater”? There is room and can delete in legend.

Done

- Add to legend: ^ Present or performed ??

Done

- Suggest: “Most of the study cores are gravity cores (GC) except for multicore 904MC, which was collected by the UiT...”

Done

- Suggest: “In 912GC, although authigenic carbonates are present based on X-ray images...”

Done

Ext Table 2 Add units of °C to table!

Done

- Add asterisk (*) to second line of table as shown;
add footnote: * Encompasses [or includes] Storfjordrenna mound

Done

- Suggest: "...water depth and are within an 8° by 2° area surrounding the Storfjordrenna mound..."

Done

- Suggest: "These measurements occurred from...mostly during spring to...measurements during winter months."

Done

- Delete last caption line, suggest: "The average and minimum temperature values do not differ much from area to area, but maximum temperatures show slightly higher variation."

Done

End Review

Reviewer #2

The manuscript reports on the first discovery of gas hydrate in the shallow parts of the Barents Sea. This in itself is very exciting and worth publication.

The authors then carry on to discuss the relevance of observed carbonate accretions in their sediment cores and conclude that seepage must have been long-term and episodic. From this they conclude that an underlying gas reservoir must release gas episodically and that the observations do not necessarily reflect climate effects. I agree with this conclusion.

The paper is of interest to the community that studies the potential effects of global warming and certainly one of the more relevant pieces of work that I have seen on this subject recently. As I am a geophysicist I cannot do justice to the geochemical modeling and I would like to ask the editor to rely on other reviewer's assessment of this part of the work. The paper is well written and illustrated and I have only one major comment and several minor suggestions for improvement. The major comment is that I would like to see a discussion if the episodicity could be caused by the functioning of the seep system on its own. Gay et al. have documented for a pockmark off Angola that the formation of carbonates leads to a deviation of gas flow. This will cause different pathways of the gas over time and may just signal episodicity while the gas is in fact coming up all the time but in different places. If you just look at three cores from the area you would not pick this up. Such a scenario is supported by your observation of small fractures. Potentially addressing this comment may change the conclusions quite a bit.

We acknowledge the reviewer's suggestion. We have added discussion (line 229-234) to address this potential process. We cannot find the paper mentioned by the reviewer but cited another publication, Marcon et al., 2013, that indicate redirection of fluid by authigenic carbonates. We agree that the gas supply beneath might never stop; pathway of gas/fluid flow changed through the time depends on the sediment texture and conduit conductivity, and note so in the revised text.

The two smaller comments: there is a weird sentence in the first paragraph (l. 58) in which you state that 5-88 % of the Arctic hydrate may dissolve due to global warming. This is just arm-waiving and ignores the fact that most of it would quickly reform hydrate. I would just remove this alarmist sentence as it is not necessary.

We agree that such numbers may not necessarily reflect the truth. We kept the reference to emphasize the point that Arctic gas hydrates are vulnerable to climate change but remove the exact numbers (Line 22-25). The new sentence does not sound alarmist, especially in the context of the rest of the manuscript which de-emphasize the gas hydrate as a "trigger" for methane release.

Further down (l. 65) you seem to say that you infer methane because you see sulfate. I think this is just an imprecise statement and you don't mean it. This should be clarified. The list of references is very complete and should be shortened. I thought there was a limit on the number of references for Nature? I noticed one problem here: you give three references for a 2 degree seasonal change of BWT, but none of these is really suitable to make the point. I hope the other references are more precise. Overall, I enjoyed reading the paper and think it is suitable for publication after minor changes.

We have clarified this statement (Line 61-63). We have also checked our reference.

Christian Berndt, June 2016

Reviewers' comments:

Reviewer #2 (Remarks to the Author):

My main comments have been addressed satisfactorily and I think the paper should be published.

I still think the number of references should be cut down from 50 to 20-30.

Christian Berndt, October 2016

Reviewer #3 (Remarks to the Author):

I was asked to review the manuscript by Hong et al. as a "third referee", awkwardly after the initial round. I also was asked if the comments by original Referee #1 had been addressed.

The MS by Hong et al. examines pore water sulfate profiles above a mound offshore Svalbard. The authors document gas hydrate in shallow sediment on the mound, and determine some unusual sulfate profiles. From the sulfate profiles, the authors suggest recently enhanced upward flow of methane. They then suggest that this is not because of a rise in bottom water temperature, but rather because of complex methane dynamics.

The basic tenets are okay, but the MS needs considerable work. Frankly, I find it difficult to read and assess. The main purpose seems to be to evaluate whether increases in bottom temperature can dissociate gas hydrate (Lines 56-62). However, the MS never really sets up the problem. As a consequence, there is considerable discussion about an intriguing problem but with major assumptions and minimal data. Even the title is odd – it suggests a switch – but no such switch is rigorously identified.

I discuss some of the problems below. Some of the comments by referee #1 were not really addressed.

Sincerely,

Gerald Dickens

Dickens Commentary

** Made before reading the rebuttal

(A) There are now many pore water sulfate profiles in regions with gas hydrate. The "kink-type" pore water profiles on the mound presented by Hong et al. are unusual, but by no means unique (e.g., Hensen et al., GCA, 2003; Fischer et al., Biogeosciences, 2012).

Most papers discussing such profiles have suggested bio-irrigation. However, as originally discussed by Schulz et al. (GCA, 1994), there are other means. These include (i) a change in sedimentation, (ii) a change in porosity, and (iii) sulfide oxidation. The present manuscript dismisses bio-irrigation on the basis of high dissolved iron concentrations (although, interestingly, there is no discussion as to whether organisms live on these mounds, which might extend processes through the upper sediment).

In any case, the other explanations have not been addressed; rather, the authors immediately jump to a change in methane flux. The authors should be able to dismiss a change in sedimentation or a change (decrease) in porosity by examining the cores. However, they should

also recognize differential compaction with gravity coring.

Ideally, they should have measured one or more dissolved species that typically have steep pore water gradients in such environments but are obviously affected by AOM (e.g., NH_4^+).

(B) There are a series of papers that have tried to model the evolution of sulfate profiles with changing methane flux. Before Hensen et al. (GCA, 2003), there was Dickens (GCA, 2001). In both these papers, the evolution of the sulfate profile was modeled over time. This aspect is mostly missing in the present paper (at least it is not shown). Instead, there is an assumed starting profile and some modern profile. A key point in previous works is that the change in pore water profiles is pretty fast.

(C) Importantly, the aforementioned papers focused on diffusive systems, where gas hydrate occurs at considerable depth (>150 m below the seafloor on Blake Ridge). The present case also using diffusion modeling – except the system is a seep, which implies significant advection!

(D) Why is the $\delta^{13}\text{C}$ of carbonate between -26 and -35 per mil? This is not discussed.

(E) The location and timing of authigenic carbonate precipitation is not clear. If the methane flux was low in the past, why do multiple layers of carbonate occur, some above the SMT? I would think very difficult to precipitate these layers in the time duration since the supposed change in methane flux.

(F) The ages discussed in the modeling are not clearly justified.

(G) Parts of the MS, such as the end conclusion (Lines 244-260), do not make full sense, because they do not appear to come from a complete framework where methane and gas hydrate are part of dynamic systems. There are both inputs and outputs of carbon to and from the ocean and biosphere that operate over time.

Some specific comments (I could make many more, but the MS needs much more work, should the authors want to make it a strong MS).

Lines 18-20: I do not fully understand this sentence. This is because changes in temperature can be part of a dynamic system that can cause episodic ventilation .

Lines 24-25: Not sure the point of this sentence. I would remove.

Lines 25-27: The authors might wish to know that this amount is probably overestimated (see Miller et al., Biogeosciences Discussion, 2016). However, the issue does not impact the present work.

Lines 27-29. Gas hydrate dissociation also has been modeled by Stranne and colleagues (G3, 2016). Crucially, though, these authors point to an issue: once free methane gas is formed from dissociated gas hydrate, it does not flow very readily, perhaps even with the generation of fractures.

Lines 65-67: See comment A.

Lines 149-151: I do not follow this without additional explanation.

Line 159: Singular verb as plural noun (operate).

Line 184: This seems too fast (see Stranne et al., 2016).

Line 185: What is "our temperature measurements" mean?

Line 218: word missing.

Figures 3 and 4 will be challenging for anyone with color vision deficiency,

In the rebuttal:

[REFEREE] This reviewer is in the awkward position of agreeing with the paper's overall hypothesis – subsurface control of upward methane migration rather than surface or near-surface thermal control – but not agreeing with the central assumption of its approach.

[AUTHORS] We have provided additional sensitivity tests and data that strengthens our initial assumptions, as detailed in lines 118-123.

*** I agree very much with the original referee comment. The root problem remains that the assumptions are high and the totality of data low. I note additional assumptions above that were not pointed out previously.

Reviewer #2 (Remarks to the Author):

My main comments have been addressed satisfactorily and I think the paper should be published. I still think the number of references should be cut down from 50 to 20-30.

Christian Berndt, October 2016

We appreciate the positive feedback and we will decrease the citation as much as possible.

Reviewer #3

I was asked to review the manuscript by Hong et al. as a “third referee”, awkwardly after the initial round. I also was asked if the comments by original Referee #1 had been addressed.

The MS by Hong et al. examines pore water sulfate profiles above a mound offshore Svalbard. The authors document gas hydrate in shallow sediment on the mound, and determine some unusual sulfate profiles. From the sulfate profiles, the authors suggest recently enhanced upward flow of methane. They then suggest that this is not because of a rise in bottom water temperature, but rather because of complex methane dynamics.

The basic tenets are okay, but the MS needs considerable work. Frankly, I find it difficult to read and assess. The main purpose seems to be to evaluate whether increases in bottom temperature can dissociate gas hydrate (Lines 56-62). However, the MS never really sets up the problem. As a consequence, there is considerable discussion about an intriguing problem but with major assumptions and minimal data. Even the title is odd – it suggests a switch – but no such switch is rigorously identified. I discuss some of the problems below. Some of the comments by referee #1 were not really addressed.

Sincerely,

Gerald Dickens

We appreciate the time from the third reviewer, Dr. Gerald Dickens, for the effort reviewing our paper. His constructive comments help improve this work. In this rebuttal letter, we will first reply to the major criticisms raised by Dr. Dickens. Our point-by-point reply to specific comments is provided afterwards. We have changed the title to “*Are Arctic shallow marine gas hydrate reservoirs sensitive to ocean warming?*”

(A) There are now many pore water sulfate profiles in regions with gas hydrate. The “kink-type” pore water profiles on the mound presented by Hong et al. are unusual, but by no means unique (e.g., Hensen et al., GCA, 2003; Fischer et al., Biogeosciences, 2012). Most papers discussing such profiles have suggested bio-irrigation. However, as originally discussed by Schulz et al. (GCA, 1994), there are other means. These include (i) a change in sedimentation, (ii) a change in porosity, and (iii) sulfide oxidation. The present manuscript dismisses bio-irrigation on the basis of high dissolved iron concentrations (although, interestingly, there is no discussion as to whether organisms live on these mounds, which might extend processes through the upper sediment). In any case, the other explanations have not been addressed; rather, the authors immediately jump to a change in methane flux. The authors should be able to dismiss a change in sedimentation or a change (decrease) in porosity by examining the cores. However, they should also recognize differential compaction with gravity coring. Ideally, they should have measured one or more dissolved species that typically have steep pore water gradients in such environments but are obviously affected by AOM (e.g., NH_4^+).

We now include an entire section for the discussion of the several interpretations of non-steady-state sulfate profiles in the literature (start from Line 113). Specifically, we discuss four scenarios in this section, as recommended by Dr. Dickens: (i) irrigation and seawater intrusion due to biological, physical, and hydrological processes; (ii) changes in sedimentation rate and porosity; (iii) reoxidation of reduced S species and; (iv) changes in methane flux. We present new porewater data of ΣHS , total alkalinity, ammonium, and calcium, magnesium from the three sites with non-steady state sulfate profiles and one site with steady-state porewater profiles (see new Figure E2). As shown by these new data, we observe similar kinks in the ΣHS , total alkalinity, calcium, and magnesium (all affected by changes in AOM rates) but not in the ammonium profiles. We obtained two ^{14}C dating results and provide a stratigraphic correlation based on XRF Zr/Rb data to show there is no anomalous sedimentation event (Figure 2) that may result in the observed sulfate profiles. Although reliable porosity measurements are not yet available, we use the chloride content in sediments from our XRF scanning as a proxy for water content (Figure 1S in the *Supplementary material*). We observed no difference in chloride ratio between the sites

with and without the non-steady state sulfate profile. In addition to these new data, to further assess the effect of these different processes on porewater profiles, we ran numerical simulations for four different scenarios that may result in the observed profiles, by considering all available data (see description in the *Supplementary material*). We conclude that these processes cannot reproduce the shape of all porewater profiles except for the case with an increasing methane flux. See the full explanation in *Supplementary material* and Lines 124-143 in the main text.

(B) There are a series of papers that have tried to model the evolution of sulfate profiles with changing methane flux. Before Hensen et al. (GCA, 2003), there was Dickens (GCA, 2001). In both these papers, the evolution of the sulfate profile was modeled over time. This aspect is mostly missing in the present paper (at least it is not shown). Instead, there is an assumed starting profile and some modern profile. A key point in previous works is that the change in pore water profiles is pretty fast.

We have amended our new Figure 3 with several snapshots showing the evolution of the profiles. The evolution can take a few decades (940GC) to a few years (911GC and 1520GC) depending on the flux of methane from deeper sediments.

Importantly, the aforementioned papers focused on diffusive systems, where gas hydrate occurs at considerable depth (>150 m below the seafloor on Blake Ridge). The present case also using diffusion modeling – except the system is a seep, which implies significant advection!

We agree that advection might be important to our system and will shorten the time required for the porewater system to evolve. We have tested advection with our comprehensive model (S5 in Figure E3). Our results show that strong advection will result in concave downward sulfate, calcium, and magnesium profiles which were not observed from the measurements (see S5 in Figure E3). We therefore conclude that whereas methane gas is clearly migrating upwards, there is no significant component of aqueous advection at our sites. A decoupling of gas and water transport (with gas dominated by advection and dissolved solutes by diffusion) have been documented in other gas hydrate regions (e.g., Torres et al., 2004, Kim et al., 2012) Our full

simulation model is consistent with such decoupling in the GHMs. Our time estimates with the assumption of diffusion-only is therefore reasonable. We have explained this in Lines 179-192.

(C) Why is the $\delta^{13}\text{C}$ of carbonate between -26 and -35 per mil? This is not discussed.

These carbon isotopic signatures are similar to the values observed from Barents Sea (Cremiere et al., 2016). We speculate the heavy carbon isotope of our authigenic carbonates is related to the thermogenic methane in the area. Thermogenic methane has been documented as a significant contribution in the Barents Sea and Svalbard region (e.g., Sahling et al., 2004; Smith et al., 2014). The source of methane in Storfjordrenna is discussed in another paper by one of the coauthor (Serov et al., in review). We did not discuss this in paper because this is not relevant to our central research question.

(D) The location and timing of authigenic carbonate precipitation is not clear. If the methane flux was low in the past, why do multiple layers of carbonate occur, some above the SMT? I would think very difficult to precipitate these layers in the time duration since the supposed change in methane flux.

As the bicarbonate produced through AOM will diffuse towards seafloor, it is likely to precipitate authigenic carbonate above SMTZ. We explained this in Lines 252-255. In the new revised text, however, we limit our inferences based on the presence of carbonate nodules to suggest a longer history of methane discharge in this area. The full discussion of carbonate formation, their composition and potential carbon cycling pathways requires a different contribution.

(E) The ages discussed in the modeling are not clearly justified.

We have obtained two ^{14}C dates from benthic foraminifera from our background sites (1522GC) (Figure 2) and use Zr/Rb ratio from XRF scanning for a stratigraphy correlation. These results have provide an age constraints for the study sites and put our inferred timing estimates through porewater modeling into a realistic framewrok. We deleted detailed time the inferences based on carbonate nodule occurrences, and limit our conclusions based on timing to modeling of pore

water data. Carbonate occurrences are mentioned at the end as potential indicators of longer time frames for the methane discharge

(F) Parts of the MS, such as the end conclusion (Lines 244-260), do not make full sense, because they do not appear to come from a complete framework where methane and gas hydrate are part of dynamic systems. There are both inputs and outputs of carbon to and from the ocean and biosphere that operate over time. Some specific comments (I could make many more, but the MS needs much more work, should the authors want to make it a strong MS).

We fully agree with reviewer's point that methane and gas hydrate should be seen as a part of the overall dynamic system. We revised the last paragraph to emphasize the overall framework with methane release and gas hydrate modulated by large scale processes (Lines 282-285). In our conclusion paragraph, we urge the view of seeing the entire system as a whole rather than over-emphasizing the small perturbations in the system, which is also important but shouldn't be the sole focus. We linked this message to one of the reviewer's paper, which assessed carbon cycling among the important carbon reservoirs, including gas hydrate, of the Earth (285-287). The conclusion of our current work emphasize such view.

Lines 18-20: I do not fully understand this sentence. This is because changes in temperature can be part of a dynamic system that can cause episodic ventilation.

We have specified the episodic ventilation of *deep reservoirs* rather than warming induced gas hydrate at *shallow* depth (Line 18-20).

Lines 24-25: Not sure the point of this sentence. I would remove.

This sentence is to emphasize hydrates can hold great amount of methane gas and justifies why gas hydrate dissociation has received such a tremendous attention in recent years.

Lines 25-27: The authors might wish to know that this amount is probably overestimated (see Miller et al., Biogeosciences Discussion, 2016). However, the issue does not impact the present work.

We are aware of this paper. The authors (Miller et al.) provide no estimates of hydrate abundance from Arctic. They reported a core transect from Siberia where no methane was found, and

conclude that the total abundance of hydrate in Arctic is overestimated. It is clear that the overall estimates of gas hydrate abundance are still uncertain, but as Dr. Dickens points out, this does not impact our conclusion and we have exceeded the citation limit (see comments by reviewer 1), we will not include the citation.

Lines 27-29. Gas hydrate dissociation also has been modeled by Stranne and colleagues (G3, 2016). Crucially, though, these authors point to an issue: once free methane gas is formed from dissociated gas hydrate, it does not flow very readily, perhaps even with the generation of fractures.

We acknowledge this information from the reviewer. However, as this is not a crucial citation for the current work, as our inferences pertain only to gas released during core recovery and formation of microfractures as indicators for the presence of gas in the sediment. We will not include the reference here, because of the limit in citations

Lines 65-67: See comment A.

See our first reply for comment A.

Lines 149-151: I do not follow this without additional explanation.

We have removed the sentences.

Line 159: Singular verb as plural noun (operate).

Corrected (Line 178)

Line 184: This seems too fast (see Stranne et al., 2016).

This part has been removed from the MS.

Line 185: What is “our temperature measurements” mean?

This part has been removed from the MS.

Line 218: word missing.

Rephrased (Line 268)

Figures 3 and 4 will be challenging for anyone with color vision deficiency,

We have changed the colors of the figures.

In the rebuttal:

[REFEREE] This reviewer is in the awkward position of agreeing with the paper’s overall hypothesis –subsurface control of upward methane migration rather than surface or near-surface thermal control – but not agreeing with the central assumption of its approach.

[AUTHORS] We have provided additional sensitivity tests and data that strengthens our initial assumptions, as detailed in lines 118-123.

*** I agree very much with the original referee comment. The root problem remains that the assumptions are high and the totality of data low. I note additional assumptions above that were not pointed out previously.

We have added a full set of porewater data to strengthen our conclusion that the non-steady-state in the sulfate profiles are due to an increase of methane flux. Our ^{14}C dating results and stratigraphic correlations establish a solid temporal frame for our study sites (see new Figure 2). The timing we estimated through modeling the porewater sulfate profiles is reasonable considering such age frame.

REVIEWERS' COMMENTS:

Reviewer #3 (Remarks to the Author):

The topic is important and interesting, and the authors have addressed all comments by three referees. Do I really like the MS? Sort of mixed views, as I still think there are problems and issues. But this is the beauty of science. If all was perfect and complete, I might as well retire on beach.

Jerry